# Accelerating Robotic Reinforcement Learning via Parameterized Action Primitives

**Murtaza Dalal**    **Deepak Pathak**[†]    **Ruslan Salakhutdinov**[†]
Carnegie Mellon University
{mdalal,dpathak,rsalakhu} @ cs.cmu.edu

## Abstract

Despite the potential of reinforcement learning (RL) for building general-purpose robotic systems, training RL agents to solve robotics tasks still remains challenging due to the difficulty of exploration in purely continuous action spaces. Addressing this problem is an active area of research with the majority of focus on improving RL methods via better optimization or more efficient exploration. An alternate but important component to consider improving is the interface of the RL algorithm with the robot. In this work, we manually specify a library of robot action primitives (RAPS), parameterized with arguments that are learned by an RL policy. These parameterized primitives are expressive, simple to implement, enable efficient exploration and can be transferred across robots, tasks and environments. We perform a thorough empirical study across challenging tasks in three distinct domains with image input and a sparse terminal reward. We find that our simple change to the action interface substantially improves both the learning efficiency and task performance irrespective of the underlying RL algorithm, significantly outperforming prior methods which learn skills from offline expert data. Code and videos at https://mihdalal.github.io/raps/

## 1 Introduction

Meaningful exploration remains a challenge for robotic reinforcement learning systems. For example, in the manipulation tasks shown in Figure 1, useful exploration might correspond to picking up and placing objects in different configurations. However, random motions in the robot's joint space will rarely, if ever, result in the robot touching the objects, let alone pick them up. Recent work, on the other hand, has demonstrated remarkable success in training RL agents to solve manipulation tasks [4, 24, 26] by sidestepping the exploration problem with careful engineering. Levine et al. [26] use densely shaped rewards, while Kalashnikov et al. [24] leverage a large scale robot infrastructure and Andrychowicz et al. [4] require training in simulation with engineered reward functions in order to transfer to the real world. In general, RL methods can be prohibitively data inefficient, require careful reward development to learn, and struggle to scale to more complex tasks without the aid of human demonstrations or carefully designed simulation setups.

An alternative view on why RL is difficult for robotics is that it requires the agent to learn both *what* to do in order to achieve the task and *how* to control the robot to execute the desired motions. For example, in the kitchen environment featured at the bottom of Figure 1, the agent would have to learn how to accurately manipulate the arm to reach different locations as well as how to grasp different objects, while also ascertaining what object it has to grasp and where to move it. Considered independently, the problems of controlling a robot arm to execute particular motions and figuring out the desired task from scalar reward feedback, then achieving it, are non-trivial. Jointly learning to solve both problems makes the task significantly more difficult.

---

[†]Equal advising

35th Conference on Neural Information Processing Systems (NeurIPS 2021).

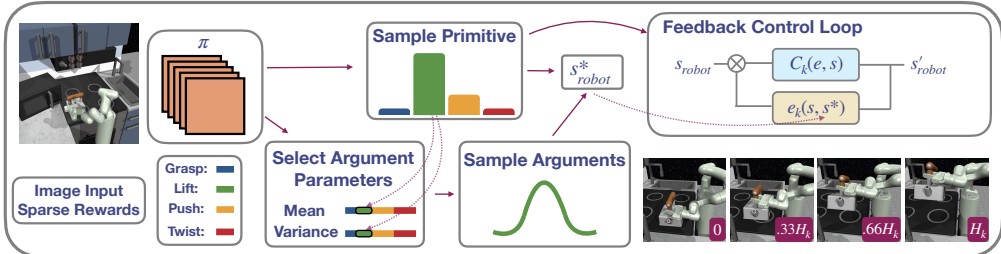

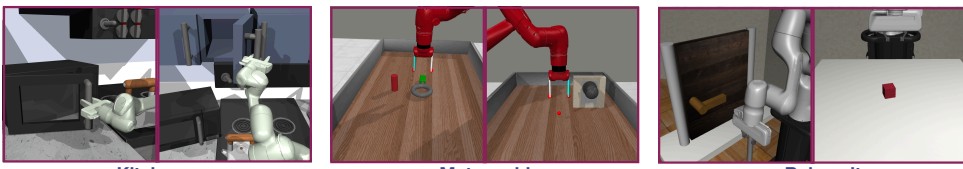

Figure 1: Visual depiction of RAPS, outlining the process of how a primitive is executed on a robot. Given an input image, the policy outputs a distribution over primitives and a distribution over all the arguments of all primitives, samples a primitive and selects its corresponding argument distribution parameters, indexed by which primitive was chosen, samples an argument from that distribution and executes a controller in a feedback loop on the robot for a fixed number of timesteps ($H_k$) to reach a new state. We show an example sequence of executing the `lift` primitive after having grasped the kettle in the Kitchen environment. The agent observes the initial (0) and final states ($H_k$) and receives a reward equal to the reward accumulated when executing the primitive. Below we visualize representative tasks from the three environment suites that we evaluate on.

In contrast to training RL agents on raw actions such as torques or delta positions, a common strategy is to decompose the agent action space into higher (i.e., *what*) and lower (i.e., *how*) level structures. A number of existing methods have focused on designing or learning this structure, from manually architecting and fine-tuning action hierarchies [14, 27, 32, 47], to organizing agent trajectories into distinct skills [3, 20, 41, 50] to more recent work on leveraging large offline datasets in order to learn skill libraries [29, 40]. While these methods have shown success in certain settings, many of them are either too sample inefficient, do not scale well to more complex domains, or lack generality due to dependence on task relevant data.

In this work, we investigate the following question: instead of learning low-level primitives, what if we were to design primitives with minimal human effort, enable their expressiveness by parameterizing them with arguments and learn to control them with a high-level policy? Such primitives have been studied extensively in task and motion planning (TAMP) literature [22] and implemented as parameterized actions [19] in RL. We apply primitive robot motions to redefine the policy-robot interface in the context of robotic reinforcement learning. These primitives include manually defined behaviors such as `lift`, `push`, `top-grasp`, and many others. The behavior of these primitives is parameterized by arguments that are the learned outputs of a policy network. For instance, `top-grasp` is parameterized by four scalar values: grasp position (x,y), how much to move down (z) and the degree to which the gripper should close. We call this application of parameterized behaviors, Robot Action Primitives for RL (RAPS). A crucial point to note is that these parameterized actions are *easy* to design, need only be defined *once* and can be *re-used* without modification across tasks.

The main contribution of this work is to support the effectiveness of RAPS via a thorough empirical evaluation across several dimensions:

- How do parameterized primitives compare to other forms of action parameterization?
- How does RAPS compare to prior methods that learn skills from offline expert data?
- Is RAPS agnostic to the underlying RL algorithm?
- Can we stitch the primitives to perform multiple complex manipulation tasks in sequence?
- Does RAPS accelerate exploration even in the absence of extrinsic rewards?

We investigate these questions across complex manipulation environments including Kitchen Suite, Metaworld and Robosuite domains. We find that a simple parameterized action based approach outperforms prior state-of-the-art by a significant margin across most of these settings[2].

---

[2]Please view our website for performance videos and links to our code: https://mihdalal.github.io/raps/

## 2 Related Work

**Higher Level Action and Policy Spaces in Robotics**     In robotics literature, decision making over primitive actions that execute well-defined behaviors has been explored in the context of task and motion planning [9, 22, 23, 43]. However, such methods are dependent on accurate state estimation pipelines to enable planning over the argument space of primitives. One advantage of using reinforcement learning methods instead is that the agent can learn to adjust its implicit state estimates through trial and error experience. Dynamic Movement Primitive and ensuing policy search approaches [11, 21, 25, 36, 37] leverage dynamical systems to learn flexible, parameterized skills, but are sensitive to hyper-parameter tuning and often limited to the behavior cloning regime. Neural Dynamic Policies [6] incorporate dynamical structure into neural network policies for RL, but evaluate in the state based regime with dense rewards, while we show that parameterized actions can enable RL agents to efficiently explore in sparse reward settings from image input.

**Hierarchical RL and Skill Learning**     Enabling RL agents to act effectively over temporally extended horizons is a longstanding research goal in the field of hierarchical RL. Prior work introduced the options framework [45], which outlines how to leverage lower level policies as actions for a higher level policy. In this framework, parameterized action primitives can be viewed as a particular type of fixed option with an initiation set that corresponds to the arguments of the primitive. Prior work on options has focused on discovering [1, 12, 41] or fine-tuning options [5, 14, 27] in addition to learning higher level policies. Many of these methods have not been extended beyond carefully engineered state based settings. More recently, research has focused on extracting useful skills from large offline datasets of interaction data ranging from unstructured interaction data [49], play [28, 29] to demonstration data [2, 35, 39, 40, 44, 46, 53]. While these methods have been shown to be successful on certain tasks, the learned skills are only relevant for the environment they are trained on. New demonstration data must be collected to use learned skills for a new robot, a new task, or even a new camera viewpoint. Since RAPS uses manually specified primitives dependent only on the robot state, RAPS can re-use the same implementation details across robots, tasks and domains.

**Parameterized Actions in RL**     The parameterized action Markov decision process (PAMDP) formalism was first introduced in Masson et al. [31], though there is a large body of earlier work in the area of hybrid discrete-continuous control, surveyed in [7, 8]. Most recent research on PAMDPs has focused on better aligning policy architectures and RL updates with the nature of parameterized actions and has largely been limited to state based domains [13, 51]. A number of papers in this area have focused on solving a simulated robot soccer domain modeled as either a single-agent [19, 31, 48] or multi-agent [15] problem. In this paper, we consider more realistic robotics tasks that involve interaction with and manipulation of common household objects. While prior work [42] has trained RL policies to select hand-designed behaviors for simultaneous execution, we instead train RL policies to leverage more expressive, *parameterized* behaviors to solve a wide variety of tasks. Closely related to this work is Chitnis et al. [10], which develops a specific architecture for training policies over parameterized actions from *state* input and sparse rewards in the context of bi-manual robotic manipulation. Our work is orthogonal in that we demonstrate that a higher level policy architecture is sufficient to solve a large suite of manipulation tasks from image input. We additionally note that there is concurrent work [34] that also applies engineered primitives in the context of RL, however, we consider learning from image input and sparse terminal rewards.

## 3 Robot Action Primitives in RL

To address the challenge of exploration and behavior learning in continuous action spaces, we decompose a desired task into the *what* (high level task) and the *how* (control motion). The *what* is handled by the *environment-centric* RL policy while the *how* is handled by a fixed, manually defined set of *agent-centric* primitives parameterized by continuous arguments. This enables the high level policy to reason about the task at a high level by choosing primitives and their arguments while leaving the low-level control to the parameterized actions themselves.

### 3.1 Background

Let the Markov decision process (MDP) be defined as $(\mathcal{S}, \mathcal{A}, \mathcal{R}(s, a, s'), \mathcal{T}(s'|s, a), p(s_0), \gamma, )$ in which $\mathcal{S}$ is the set of true states, $\mathcal{A}$ is the set of possible actions, $\mathcal{R}(s, a, s')$ is the reward function, $\mathcal{T}(s'|s, a)$ is the transition probability distribution, $p(s_0)$ defines the initial state distribution, and $\gamma$ is the discount factor. The agent executes actions in the environment using a policy $\pi(a|s)$ with a

corresponding trajectory distribution $p(\tau = (s_0, a_0, ...a_{t-1}, s_T)) = p(s_0)\Pi_t \pi(a_t|s_t)\mathcal{T}(s_{t+1}|s_t, a_t)$. The goal of the RL agent is to maximize the expected sum of rewards with respect to the policy: $\mathbb{E}_{s_0, a_0, ...a_{t-1}, s_T, \sim p(\tau)}\left[\sum_t \gamma^t \mathcal{R}(s_t, a_t)\right]$. In the case of vision-based RL, the setup is now a partially observed Markov decision process (POMDP); we have access to the true state via image observations. In this case, we include an observation space $\mathcal{O}$ which corresponds to the set of visual observations that the environment may emit, an observation model $p(o|s)$ which defines the probability of emission and policy $\pi(a|o)$ which operates over observations. In this work, we consider various modifications to the action space $\mathcal{A}$ while keeping all other components of the MDP or POMDP the same.

### 3.2 Parameterized Action Primitives

We now describe the specific nature of our parameterized primitives and how they can be integrated into RL algorithms (see Figure 1 for an end-to-end visualization of the method). In a library of $K$ primitives, the k-th primitive is a function $f_k(s, \text{args})$ that executes a controller $C_k$ on a robot for a fixed horizon $H_k$, $s$ is the robot state and args is the value of the arguments passed to $f_k$. args is used to compute a target robot state $s^*$ and then $C_k$ is used to drive $s$ to $s^*$. A primitive dependent error metric $e_k(s, s^*)$ determines the trajectory $C_k$ takes to reach $s^*$. $C_k$ is a general purpose state reaching controller, e.g. an end-effector or joint position controller; we assume access to such a controller for each robot and it is straightforward to define and tune if not provided. In this case, the same primitive implementation can be re-used across any robot. In this setup, the choice of controller, error metric and method to compute $s^*$ define the behavior of the primitive motion, how it uniquely forms a movement in space. We refer to Procedure 1 for a general outline of a parameterized primitive. To summarize, each skill is a feedback control loop with end-effector low level actions. The input arguments are used to define a target state to achieve and the primitive executes a loop to drive the error between the robot state and the target robot state to zero.

As an example, consider the "lifting" primitive, which simply involves lifting the robot arm upward. For this action, args is the amount to lift the robot arm, e.g. by 20cm., the robot state for this primitive is the robot end-effector position, k is the index of the lifting primitive in the library, $C_k$ is an end-effector controller, $e_k(s, s^*) = s^* - s$, and $H_k$ is the end-effector controller horizon, which in our setting ranges from 100-300. The target position $s^*$ is computed as $s + [0, 0, \text{args}]$. $f$ moves the robot arm for $H_k$ steps, driving $s$ towards $s^*$. The other primitives are defined in a similar manner; see the appendix for a precise description of each primitive we define.

Robot action primitives are a function of the robot state, not the world state. The primitives function by reaching set points of the robot state as directed by the policy, hence they are *agent-centric*. This design makes primitives agnostic to camera view, visual distractors and even the underlying environment itself. The RL policy, on the other hand, is *environment centric*: it chooses the primitive and appropriate arguments based on environment observations in order to

---

**Procedure 1** Parameterized Action Primitive

**Input:** primitive dependent argument vector args, primitive index $k$, robot state $s$
1: compute $s^*(\text{args}, s)$
2: **for** $i = 1, ..., H_k$ low-level steps **do**
3:     $e_i = e_k(s_i, s^*)$         ▷ compute state error
4:     $a_i = C_k(e_i, s_i)$          ▷ compute torques
5:     execute $a_i$ on robot
6: **end for**

---

best achieve the task. A key advantage of this decomposition is that the policy no longer has to learn *how* to move the robot and can focus directly on *what* it needs to do. Meanwhile, the low-level control need not be perfect because the policy can account for most discrepancies using the arguments.

One issue with using a fixed library of primitives is that it cannot define all possible robot motions. As a result, we include a dummy primitive that corresponds to the raw action space. The dummy primitive directly takes in a delta position and then tries to achieve it by taking a fixed number of steps. This does not entirely resolve the issue as the dummy primitive operates on the high level horizon for $H_k$ steps when called. Since the primitive is given a fixed goal for $H_k$ steps, it is less expressive than a feedback policy that could provide a changing argument at every low-level step. For example, if the task is to move in a circle, the dummy primitive with a fixed argument could not provide a target state that would directly result in the desired motion without resorting to a significant number of higher level actions, while a feedback policy could iteratively update the target state to produce a smooth motion in a circle. Therefore, it cannot execute every trajectory that a lower level policy could; however, the primitive library as a whole performs well in practice.

In order to integrate these parameterized actions into the RL setting, we modify the action space of a standard RL environment to involve two operations at each time step: (a) choose a primitive

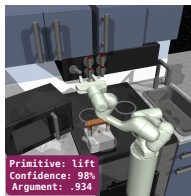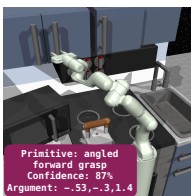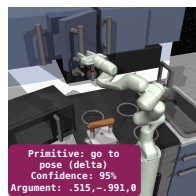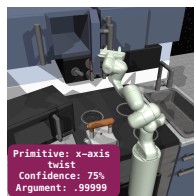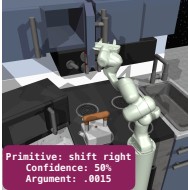

Figure 2: We visualize an execution of an RL agent trained to solve a cabinet opening task from sparse rewards using robot action primitives. At each time-step, we display the primitive chosen, the policy's confidence in the action choice and the corresponding argument passed to the primitive in the bottom left corner.

out of a fixed library (b) output its arguments. As in Chitnis et al. [10], the policy network outputs a distribution over one-hot vectors defining which primitive to use as well as a distribution over all of the arguments for all of the primitives, a design choice which enables the policy network to have a fixed output dimension. After the policy samples an action, the chosen parameterized action and its corresponding arguments are indexed from the action and passed to the environment. The environment selects the appropriate primitive function $f$ and executes the primitive on the robot with the appropriate arguments. After the primitive completes executing, the final observation and sum of intermediate rewards during the execution of the primitive are returned by the environment. We do so to ensure if the task is achieved mid primitive execution, the action is still labelled successful.

We describe a concrete example to ground the description of our framework. If we have 10 primitives with 3 arguments each, the higher level policy network outputs 30 dimensional mean and standard deviation vectors from which we sample a 30 dimensional argument vector. It also outputs a 10 dimensional logit vector from which we sample a 10 dimensional one-hot vector. Therefore in total, our action space would be 40 dimensional. The environment takes in the 40 dimensional vector and selects the appropriate argument (3-dimensional vector) from the argument vector based on the one-hot vector over primitives and executes the corresponding primitive in the environment. Using this policy architecture and primitive execution format, we train standard RL agents to solve manipulation tasks from sparse rewards. See Figure 2 for a visualization of a full trajectory of a policy solving a hinge cabinet opening task in the Kitchen Suite with RAPS.

## 4  Experimental Setup

In order to perform a robust evaluation of robot action primitives and prior work, we select a set of challenging robotic control tasks, define our environmental setup, propose appropriate metrics for evaluating different action spaces, and summarize our baselines for comparison.

**Tasks and Environments**:    We evaluate RAPS on three simulated domains: Metaworld [17], Kitchen [52] and Robosuite [54], containing 16 tasks with varying levels of difficulty, realism and task diversity (see the bottom half of Fig. 1). We use the Kitchen environment because it contains seven different subtasks within a single setting, contains human demonstration data useful for training learned skills and contains tasks that require chaining together up to four subtasks to solve. In particular, learning such temporally-extended behavior is challenging [2, 17, 35]. Next, we evaluate on the Metaworld benchmark suite due to its wide range of manipulation tasks and established presence in the RL community. We select a subset of tasks from Metaworld (see appendix) with different solution behaviors to robustly evaluate the impact of primitives on RL. Finally, one limitation of the two previous domains is that the underlying end-effector control is implemented via a simulation constraint as opposed to true position control by applying torques to the robot. In order to evaluate if primitives would scale to more realistic learning setups, we test on Robosuite, a benchmark of robotic manipulation tasks which emphasizes realistic simulation and control. We select the block lifting and door opening environments which have been demonstrated to be solvable in prior work [54]. We refer the reader to the appendix for a detailed description of each environment.

**Sparse Reward and Image Observations**    We modify each task to use the environment success metric as a sparse reward which returns 1 when the task is achieved, and 0 otherwise. We do so in order to establish a more realistic and difficult exploration setting than dense rewards which require significant engineering effort and true state information to compute. Additionally, we plot all results against the mean task success rate since it is a directly interpretable measure of the agent's performance. We run each method using visual input as we wish to bring our evaluation setting closer

to real world setups. The higher level policy, primitives and baseline methods are not provided access to the world state, only camera observations and robot state depending on the action.

**Evaluation Metrics**     One challenge when evaluating hierarchical action spaces such as RAPS alongside a variety of different learned skills and action parameterizations, is that of defining a fair and meaningful definition of sample efficiency. We could define one sample to be a forward pass through the RL policy. For low-level actions this is exactly the sample efficiency, for higher level actions this only measures how often the policy network makes decisions, which favors actions with a large number of low-level actions without regard for controller run-time cost, which can be significant. Alternatively, we could define one sample to be a single low-level action output by a low-level controller. This metric would accurately determine how often the robot itself acts in the world, but it can make high level actions appear deceptively inefficient. Higher level actions execute far fewer forward passes of the policy in each episode which can result in faster execution on a robot when operating over visual observations, a key point low-level sample efficiency fails to account for. We experimentally verify this point by running RAPS and raw actions on a real xArm 6 robot with visual RL and finding that RAPS executes each trajectory **32x** times faster than raw actions. We additionally verify that RAPS is efficient with respect to low level steps in Figure 4.

To ensure fair comparison across methods, we instead propose to perform evaluations with respect to two metrics, namely, (a) **Wall-clock Time**: the amount of total time it takes to train the agent to solve the task, both interaction time and time spent updating the agent, and (b) **Training Steps**: the number of gradient steps taken with a fixed batch size. Wall clock time is not inherently tied to the action space and provides an interpretable number for how long it takes for the agent to learn the task. To ensure consistency, we evaluate all methods on a single RTX 2080 GPU with 10 CPUs and 50GB of memory. However, this metric is not sufficient since there are several possible factors that can influence wall clock time which can be difficult to disambiguate, such as the effect of external processes, low-level controller execution speed, and implementation dependent details. As a result, we additionally compare methods based on the number of training steps, a proxy for data efficiency. The number of network updates is only a function of the data; it is independent of the action space, machine and simulator, making it a non-transient metric for evaluation. The combination of the two metrics provides a holistic method of comparing the performance of different action spaces and skills operating on varying frequencies and horizons.

**Baselines**     The simplest baseline we consider is the default action space of the environment, which we denote as **Raw Actions**. One way to improve upon the raw action space is to train a policy to output the parameters of the underlying controller alongside the actual input commands. This baseline, **VICES** [30], enables the agent to tune the controller automatically depending on the task. Alternatively, one can use unsupervised skill extraction to generate higher level actions which can be leveraged by downstream RL. We evaluate one such method, **Dyn-E** [49], which trains an observation and action representation from random policy data such that the subsequent state is predictable from the embeddings of the previous observation and action. A more data-driven approach to learning skills involves organizing demonstration data into a latent skill space. Since the dataset is guaranteed to contain meaningful behaviors, it is more likely that the extracted skills will be useful for downstream tasks. We compare against **SPIRL** [35], a method that ingests a demonstration dataset to train a fixed length skill VAE $z = e(a_{1:H}), a_{1:H} = d(z)$ and prior over skills $p(z|s)$, which is used to guide downstream RL. Additionally, we compare against **PARROT** [44], which trains an observation conditioned flow model on an offline dataset to map from the raw action space to a latent action space. In the next section, we demonstrate the performance of our RAPS against these methods across a diverse set of sparse reward manipulation tasks.

## 5   Experimental Evaluation of RAPS

We evaluate the efficacy of RAPS on three different settings: single task reinforcement learning across Kitchen, Metaworld and Robosuite, as well as hierarchical control and unsupervised exploration in the Kitchen environment. We observe across all evaluated settings, RAPS is robust, efficient and performant, in direct contrast to a wide variety of learned skills and action parameterizations.

---

[3]In all of our results, each plot shows a 95% confidence interval of the mean performance across three seeds.

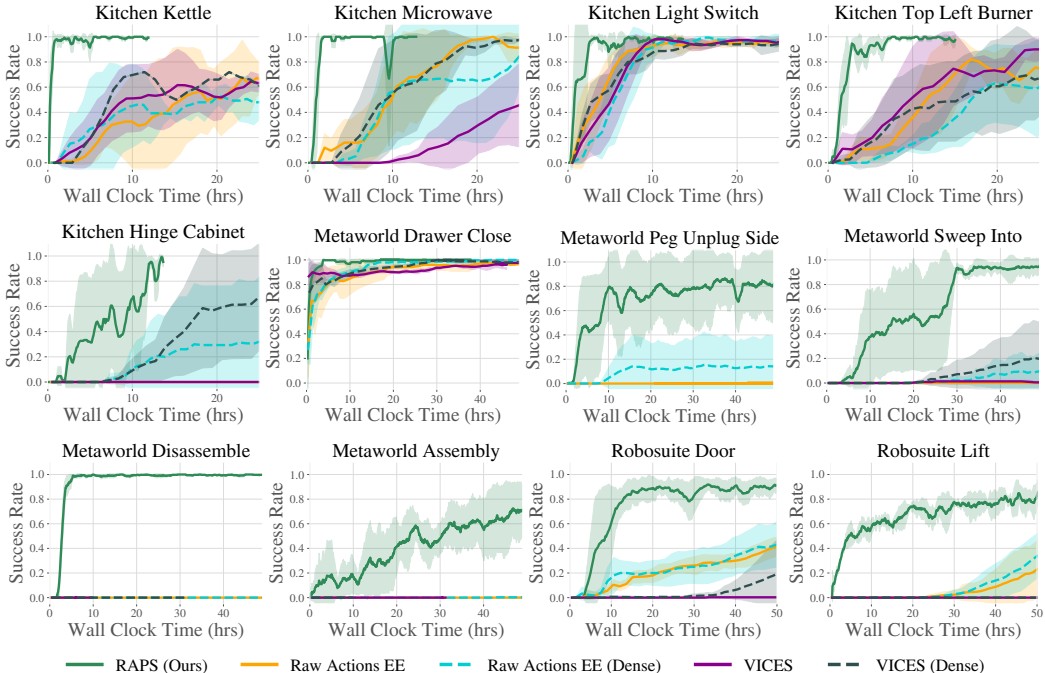

Figure 3: Comparison of various action parameterizations and RAPS across all three environment suites[3] using Dreamer as the underlying RL algorithm. RAPS (green), with sparse rewards, is able to significantly outperform all baselines, particularly on the more challenging tasks, even when they are augmented with dense reward. See the appendix for remaining plots on the `slide-cabinet` and `soccer-v2` tasks.

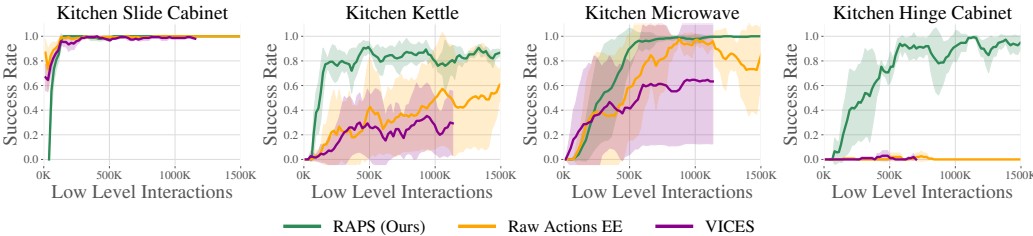

Figure 4: In the Kitchen environment suite, we run comparisons logging the number of low level interactions of RAPS, Raw actions and VICES. While the methods appear closer in efficiency with respect to low-level actions, RAPS still maintains the best performance across every task. We note that on a real robot, RAPS runs significantly faster than the raw action space in terms of wall-clock time.

## 5.1 Accelerating Single Task RL using RAPS

In this section, we evaluate the performance of RAPS against fixed and variable transformations of the lower-level action space as well as state of the art unsupervised skill extraction from demonstrations. Due to space constraints, we show performance against the number of training steps in the appendix.

**Action Parameterizations** We compare RAPS against Raw Actions and VICES using Dreamer [18] as the underlying algorithm across all three environment suites in Figure 3. Since we observe weak performance on the default action space of Kitchen, joint velocity control, we instead modify the suite to use 6DOF end-effector control for both raw actions and VICES. We find Raw Actions and VICES are able to make progress on a number of tasks across all three domains, but struggle to execute the fine-grained manipulation required to solve more difficult environments such as `hinge-cabinet`, `assembly-v2` and `disassembly-v2`. The latter two environments are not solved by Raw Actions or VICES even when they are provided dense rewards. In contrast, RAPS is able to quickly solve every task from sparse rewards.

On the kitchen environment, from sparse rewards, no prior method makes progress on the hardest manipulation task: grasping the hinge cabinet and pulling it open to 90 degrees, while RAPS is able

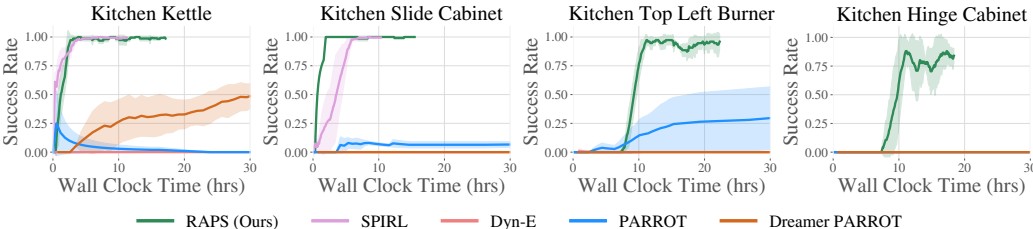

Figure 5: Comparison of RAPS and skill learning methods on the Kitchen domain using SAC as the underlying RL algorithm. While SPIRL and PARROT are competitive or even improve upon RAPS's performance on easier tasks, only RAPS (green) is able to solve `top-left-burner` and `hinge-cabinet`.

| RL Algorithm | Kettle | | Slide Cabinet | | Light Switch | | Microwave | | Top Burner | | Hinge Cabinet | |
|---|---|---|---|---|---|---|---|---|---|---|---|---|
| | Raw | RAPS | Raw | RAPS | Raw | RAPS | Raw | RAPS | Raw | RAPS | Raw | RAPS |
| Dreamer | 0.8 | **.93** | 1.0 | 1.0 | 1.0 | 1.0 | .53 | **0.8** | .93 | **1.0** | 0.0 | **1.0** |
| SAC | .33 | **0.8** | .67 | **1.0** | **.86** | .67 | .33 | **1.0** | .33 | **1.0** | 0.0 | **1.0** |
| PPO | .33 | **1.0** | .66 | **1.0** | .27 | **1.0** | 0.0 | **.66** | .27 | **1.0** | 0.0 | **1.0** |

Table 1: Evaluation of RAPS across RL algorithms (Dreamer, PPO, SAC) on Kitchen. We report the final success rate of each method on five evaluation trials trained over three seeds from sparse rewards. While raw action performance (left entry) varies significantly across RL algorithms, RAPS (right entry) is able to achieve high success rates on *every* task with *every* RL algorithm.

to quickly learn to solve the task. In the Metaworld domain, `peg-unplug-side-v2`, `assembly-v2` and `disassembly-v2` are difficult environments which present a challenge to even dense reward state based RL [52]. However, RAPS is able to solve all three tasks with *sparse rewards* directly from image input. We additionally include a comparison of RAPS against Raw Actions on all 50 Metaworld tasks with final performance shown in Figure 6 as well as the full learning performance in the Appendix. RAPS is able to learn to solve or make progress on **43 out of 50** tasks purely from sparse rewards. Finally, in the Robosuite domain, by leveraging robot action primitives, we are able to learn to solve the tasks more rapidly than raw actions or VICES, with respect to wall-clock time and number of training steps, demonstrating that RAPS scales to more realistic robotic controllers.

**Offline Learned Skills**     An alternative point of comparison is to leverage offline data to learn skills and run downstream RL. We train SPIRL and PARROT from images using the kitchen demonstration datasets in D4RL [16], and Dyn-E with random interaction data. We run all agents with SAC as the underlying RL algorithm and extract learned skills using joint velocity control, the type of action present in the demonstrations. See Figure 5 for the comparison of RAPS against learned skills. Dyn-E is unable to make progress across any of the domains due to the difficulty of extracting useful skills from highly unstructured interaction data. In contrast, SPIRL and PARROT manage to leverage demonstration data to extract useful skills; they are competitive or even improve upon RAPS on the easier tasks such as `microwave` and `kettle`, but struggle to make progress on the more difficult tasks in the suite. PARROT, in particular, exhibits a great deal of variance across tasks, especially with SAC, so we include results using Dreamer as well. We note that both SPIRL and PARROT are limited by the tasks which are present in the demonstration dataset and unable to generalize their extracted skills to other tasks in the same environment or other domains. In contrast, parameterized primitives are able to solve *all* the kitchen tasks and are re-used across domains as shown in Figure 3.

**Generalization to different RL algorithms**     A generic set of skills should maintain performance regardless of the underlying RL algorithm. In this section, we evaluate the performance of RAPS against Raw Actions on three types of RL algorithms: model based (Dreamer), off-policy model free (SAC) and on-policy model free (PPO) on the Kitchen tasks. We use the end-effector version of raw actions as our point of comparison on these tasks. As seen in Table 1, unlike raw actions, RAPS is largely agnostic to the underlying RL algorithm.

## 5.2   Enabling Hierarchical Control via RAPS

We next apply RAPS to a more complex setting: sequential RL, in which the agent must learn to solve multiple subtasks within a single episode, as opposed to one task. We evaluate on the Kitchen Multi-Task environments and plot performance across SAC, Dreamer, and PPO in Figure 7.

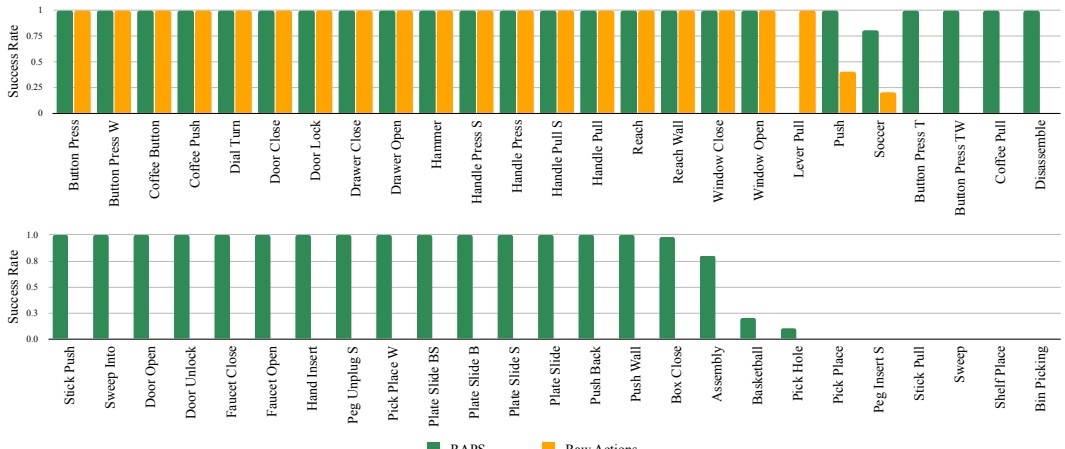

Figure 6: Final performance results for single task RL on the Metaworld domain after 3 days of training using the Dreamer base algorithm. RAPS is able to successfully learn most tasks, solving 43 out of 50 tasks while Raw Actions is only able to solve 21 tasks.

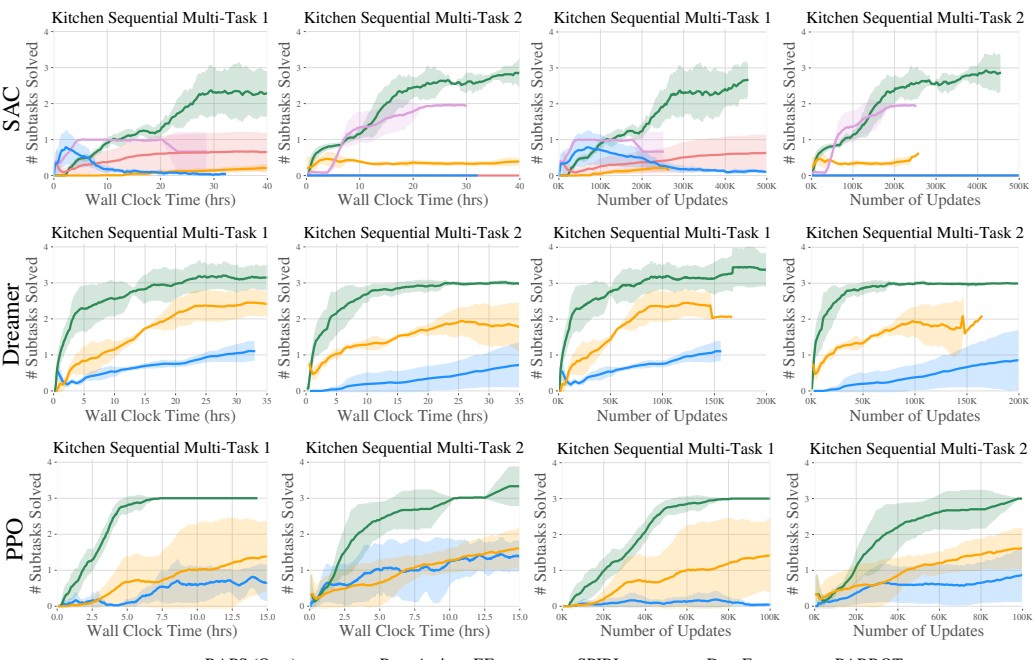

Figure 7: Learning performance of RAPS on sequential multi-task RL. Each row plots a different base RL algorithm (SAC, Dreamer, PPO) while the first two columns plot the two multi-task environment results against wall-clock time and the next two columns plot against number of updates, i.e. training steps. RAPS consistently solves at least three out of four subtasks while prior methods generally fail to make progress beyond one or two.

Raw Actions prove to be a strong baseline, eventually solving close to three subtasks on average, while requiring significantly more wall-clock time and training steps. SPIRL initially shows strong performance but after solving one to two subtasks it then plateaus and fails to improve. PARROT is less efficient than SPIRL but also able to make progress on up to two subtasks, though it exhibits a great deal of sensitivity to the underlying RL algorithm. For both of the offline skill learning methods, they struggle to solve any of the subtasks outside of `kettle`, `microwave`, and `slide-cabinet` which are encompassed in the demonstration dataset. Meanwhile, with RAPS, across all three base RL algorithms, we observe that the agents are able to leverage the primitive library to rapidly solve three out of four subtasks and continue to improve. This result demonstrates that RAPS can elicit significant gains in hierarchical RL performance through its improved exploratory behavior.

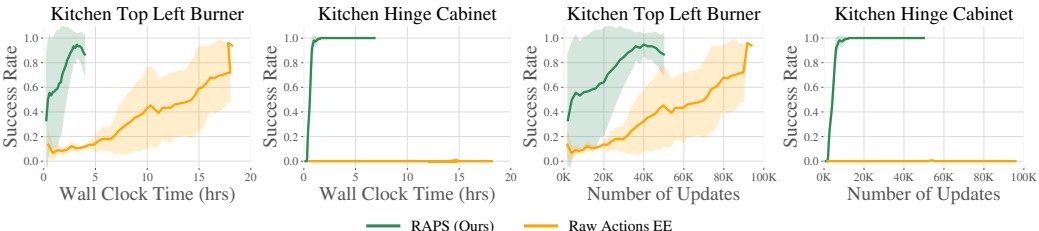

Figure 8: RAPS significantly outperforms raw actions in terms of total wall clock time and number of updates when fine-tuning initialized from reward free exploration.

## 5.3 Leveraging RAPS to enable efficient unsupervised exploration

In many settings, sparse rewards themselves can be hard to come by. Ideally, we would be able to train robot without train time task rewards for large periods of time and fine-tune to solve new tasks with only a few supervised labels. We use the kitchen environment to test the efficacy of primitives on the task of unsupervised exploration. We run an unsupervised exploration algorithm, Plan2explore [38], for a fixed number of steps to learn a world model, and then fine-tune the model and train a policy using Dreamer to solve specific tasks. We plot the results in Figure 8 on the `top-left-burner` and `hinge-cabinet` tasks. RAPS enables the agent to learn an effective world model that results in rapid learning of both tasks, requiring only **1 hour of fine-tuning** to solve the `hinge-cabinet` task. Meanwhile, the world model learned by exploring with raw actions is unable to quickly finetune as quickly. We draw two conclusions from these results, a) RAPS enables more efficient exploration than raw actions, b) RAPS facilitates efficient model fitting, resulting in rapid fine-tuning.

## 6  Discussion

**Limitations and Future Work**     While we demonstrate that RAPS is effective at solving a diverse array of manipulation tasks from visual input, there are several limitations that future work would need to address. One issue to consider is that of dynamic, fluid motions. Currently, once a primitive begins executing, it will not stop until its horizon is complete, which prevents dynamic behavior that a feedback policy on the raw action space could achieve. In the context of RAPS, integrating the parameterization and environment agnostic properties of robot action primitives with standard feedback policies could be one way to scale RAPS to more dynamic tasks. Another potential concern is that of expressivity: the set of primitives we consider in this work cannot express all possible motions that robot might need to execute. As discussed in Section 3, we do combine the base actions with primitives via a dummy primitive so that the policy can fall back to default action space if necessary. Future work could improve upon our simple solution. Finally, more complicated robot morphologies may require significant domain knowledge in order to design primitive behaviors. In this setting, we believe that learned skills with the agent-centric structure of robot action primitives could be an effective way to balance between the difficulty of learning policies to control complex robot morphologies [4, 33] and the time needed to manually define primitives.

**Conclusion**     In this work we present an extensive evaluation of RAPS, which leverages parameterized actions to learn high level policies that can quickly solve robotics tasks across three different environment suites. We show that standard methods of re-parameterizing the action space and learning skills from demonstrations are environment and domain dependent. In many cases, prior methods are unable to match the performance of robot action primitives. While primitives are not a general solution to every task, their success across a wide range of environments illustrates the utility of incorporating an agent-centric structure into the robot action space. Given the effectiveness of simple parameterized action primitives, a promising direction to further investigate would be how to best incorporate agent-centric structure into both learned and manually defined skills and attempt to get the best of both worlds in order to improve the interface of RL algorithms with robots.

**Acknowledgments**     We thank Shikhar Bahl, Ben Eyesenbach, Aravind Sivakumar, Rishi Veerapaneni, Russell Mendonca and Paul Liang for feedback on early drafts of this paper. This work was supported in part by NSF IIS1763562, NSF IIS-2024594, and ONR Grant N000141812861, and the US Army. Additionally, MD is supported by the NSF Graduate Fellowship. We would also like to acknowledge NVIDIA's GPU support.

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
