# A  Additional Experimental Results

**Cross Robot Transfer**    Robot action primitives are agnostic to the exact geometry of the underlying robot, provided the robot is a manipulator arm. As a result, one could plausibly ask the question: is it possible to train RAPS on one robot and evaluate on a morphologically different robot for the same task? In order to answer this question, we train a higher level policy over RAPS from visual input to solve the door opening task in Robosuite using the xARM 7. We then directly transfer this policy (zero-shot) to an xARM 6 robot. The transferred policy is able to achieve **100%** success rate on the door opening task with the 6DOF robot while trained on a 7DOF robot. To our knowledge such as result has not been shown before.

**Comparison against Dynamic Motion Primitives**    As noted in the related works section, Dynamic Motion Primitives (DMP) are an alternative skill formulation that is common robotics literature. We compared RAPS with the latest state-of-the-art work that incorporates DMPs with Deep RL: Neural Dynamic Policies [6]. As seen in Figure 16, across nearly every task in the Kitchen suite, RAPS outperforms NDP from visual input just as it outperforms all prior skill learning methods as well.

**Real Robot Timing Results**    To experimentally verify that RAPS runs faster than raw actions in the real wrold, we ran a randomly initialized deep RL agent with end-effector control and RAPS on a real xArm 6 robot and averaged the times of running ten trajectories. Each primitive ran 200 low-level actions with a path length of five high level actions, while the low-level path length was 500. Note that RAPS has double the number of low-level actions of the raw action space within a single trajectory. With raw actions, each episode took 16.49 seconds while with RAPS, each episode lasted an average of 0.51 seconds, a **32x** speed up.

# B  Ablations

**Primitive Usage Experiments**    We run an ablation to measure how often RAPS uses each primitive. In Figure 12, we log the number of times each primitive is called at test time, averaged across all of the kitchen environments. It is clear from the figure that even at convergence, each primitive is called a non-zero amount of times, so each primitive is useful for some task. However, there are two primitives that are favored across all the tasks, `move-delta-ee-pose` and `angled-xy-grasp`. This is not surprising as these two primitives are easily applicable to many tasks. We evaluate the number of unique primitives selected by the test time policy over time (within a single episode) in Figure 13 and note that it converges to about 2.69. To ground this number, the path length for these tasks is 5. This means that on most tasks, the higher level policy ends up repeatedly applying certain primitives in order to achieve the task.

**Evaluating the Dummy Primitive**    The dummy primitive is one of the two most used primitives (also known as move delta ee pose), the other being angled xy grasp (also known as angled forward grasp in the appendix). One question that may arise is: How useful is the dummy primitive? We run an experiment with and without the dummy primitive in order to evaluate its impact, and find that the dummy primitive improves performance significantly. Based on the results in Figure 14, hand-designed primitives are not always sufficient to solve the task.

**Using a 6DOF Control Dummy Primitive**    The dummy primitive uses 3DOF (end-effector position) control in the experiments in the main paper, but we could just as easily do 6DOF control if desired. In fact, we ablate this exact design choice. If we change the dummy primitive to achieve any full 6-DOF pose (end-effector position as well as orientation expressed in roll-pitch-yaw), the overall performance of RAPS does not change. We plot the results of running RAPS on the Kitchen tasks against RAPS with a 6DOF dummy primitive in Figure 15 and find that the performance is largely the same.

# C  Environments

We provide detailed descriptions of each environment suite and the specific tasks each suite contains. All environments use the MuJoCo simulator [51].

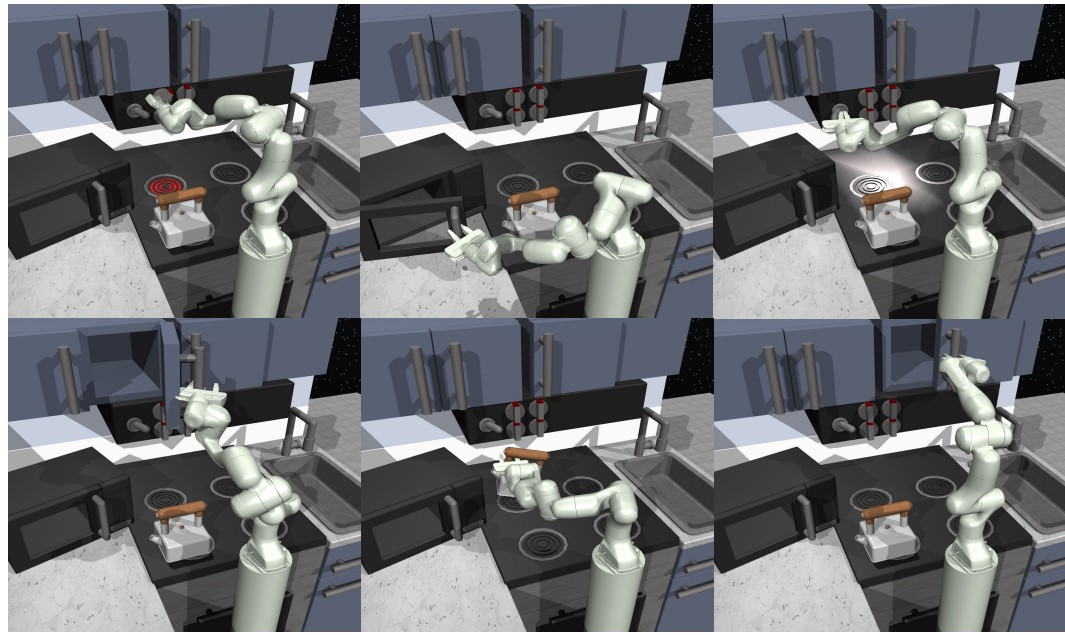

Figure 9: Visual depiction of the Kitchen environment; all tasks are contained within the same setup. Each image depicts the solution of one of the tasks, we omit the bottom burner task as it is the goal is the same as the top burner task, just with a different dial to turn. For the top row from the left: `top-left-burner`, `microwave`, `light-switch`. For the bottom row from the left: `hinge-cabinet`, `kettle`, `slide cabinet`.

## C.1 Kitchen

The Kitchen suite, introduced in [17], involves a set of different tasks in a kitchen setup with a single Franka Panda arm as visualized in Figure 9. This domain contains 7 subtasks: `slide-cabinet` (shift right-side cabinet to the right), `microwave` (open the microwave door), `kettle` (place the kettle on the back burner), `hinge-cabinet` (open the hinge cabinet), `top-left-burner` (rotate the top stove dial), `bottom-left-burner` (rotate the bottom stove dial), and `light-switch` (flick the light switch to the left). The tasks are all defined in terms of a sparse reward, in which $+1$ reward is received when the norm of the joint position (qpos in MuJoCo) of the object is within .3 of the desired goal location and 0 otherwise. See the appendix of the RPL [17] paper for the exact definition of the sparse and the dense reward functions in the kitchen environment. Since the rewards are defined simply in terms of distance of object to goal, the agent does not have to execute interpretable behavior in order to solve the task. For example, to solve the burner task, it is possible to push it to the right setting without grasping and turning it. The low level action space for this suite uses 6DOF end-effector control along with grasp control; we implement the primitives using this action space.

For the sequential multi-task version of the environment, in a single episode, the goal is to complete four different subtasks. The agent receives reward once per sub-task completed with a maximum episode return (sum of rewards) of 4. In our case, we split the 7 tasks in the environment into two multi-task environments which are roughly split on difficulty. We define the two multi-task environments in the kitchen setup: `Kitchen Multitask 1` which contains `microwave`, `kettle`, `light-switch` and `top-left-burner` while `Kitchen Multitask 2` contains the `hinge-cabinet`, `slide-cabinet`, `bottom-left-burner` and `light-switch`. As mentioned in the experiments section, RL trained on joint velocity control is unable to solve almost any of the single task environments using image input from sparse rewards. Instead, we modify the environment to use 6DOF delta position control by adding a mocap constraint as implemented in Metaworld [57].

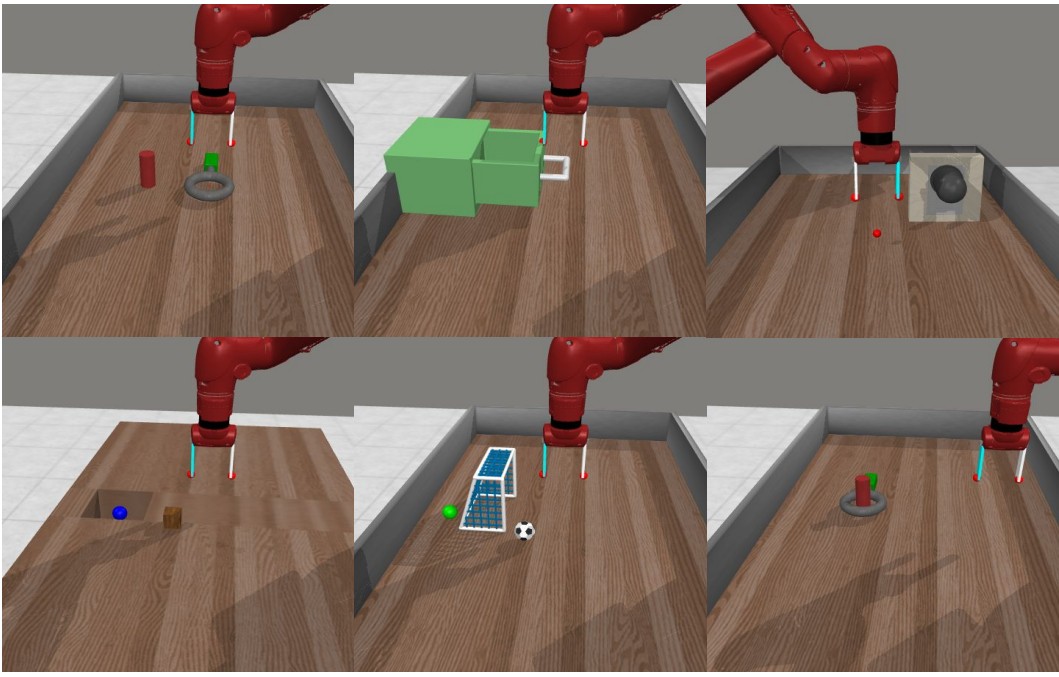

Figure 10: Visual depiction of the Metaworld environment suite. For the top row from the left: `assembly-v2`, `drawer-close-v2`, `peg-unplug-side-v2`. For the bottom row from the left: `sweep-into-v2`, `soccer-v2`, `disassemble-v2`.

## C.2 Metaworld

Metaworld [57] consists of 50 different manipulation environments in which a simulated Sawyer Rethink robot is charged with solving tasks such as faucet opening/closing, pick and place, assembly/disassembly and many others. Due to computational considerations, we selected 6 tasks which range from easy to difficult: `drawer-close-v2` (push the drawer closed), `hand-insert-v2` (place the hand inside the hole), `soccer-v2` (hit the soccer ball to a specific location in the goal box), `sweep-into-v2` (push the block into the hole), `assembly-v2` (grasp the nut and place over the thin block), and `disassembly-v2` (grasp the nut and remove from the thin block).

In Metaworld, the raw actions are delta positions, while the end-effector orientation remains fixed. For fairness, we disabled the use of any rotation primitives for this suite. Metaworld has a hand designed dense reward per task which enables efficient learning, but is unrealistic for the real world in which it can be challenging to design dense rewards without access to the true state of the world. Instead, for more realistic evaluation, we run all methods with a sparse reward which uses the success metric emitted by the environment itself. The low level action space for these environments uses 3DOF end-effector control along with grasp control; we implement the primitives using this action space.

We run the environments in single task mode, meaning the target positions remain the same across experiments, in order to evaluate the basic effectiveness of RL across action spaces. This functionality is provided in the latest release of Metaworld. Additionally, we use the V2 versions of the tasks after correspondence with the current maintainers of the benchmark. The V2 environments have a more realistic visual appearance, improved reward functions and are now the primarily supported environments in Metaworld. See Figure 10 for a visualization of the Metaworld tasks.

## C.3 Robosuite

Robosuite is a benchmark of robotic manipulation tasks which emphasizes realistic simulation and control while containing several tasks existing RL algorithms struggle to solve, even when provided state based information and dense rewards. This suite contains a torque based end-effector position control implementation, Operational Space Control [26]. We select the `lift` and `door` tasks for

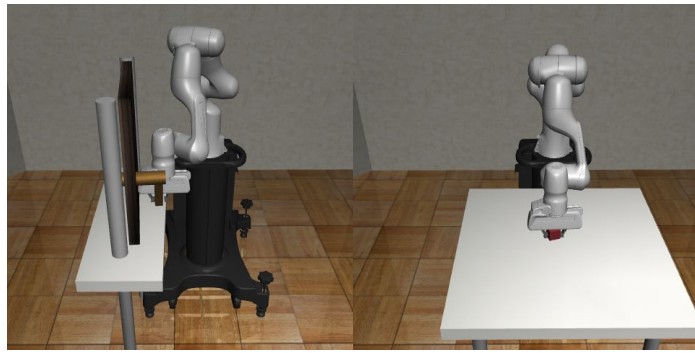

Figure 11: Visual depiction of the Robosuite environments. On the left we have the door opening task, and on the right we have the block lifting task.

evaluation, which we visualize in Figure 11. The lifting task involves accurately grasping a small red block and lifting it to a set height. The door task involves grasping the door handle, pushing it down to unlock it and pulling it open to a set position. These tasks contain initial state randomization; at each reset the position of the block or door is randomized within a small range. This property makes the Robosuite tasks more challenging than Kitchen and Metaworld, both of which are deterministic environments. For this environment, sparse rewards were already defined so we directly use them in our experiments. We made several changes to these environments to improve learning performance of the baselines as well as RAPS. Specifically, we included a workspace limit in a large area around the object, which improves exploration in the case of sparse rewards. For the lifting task, we increased the frequency of the default OSC controller to 40Hz from 20Hz, while for the door opening task we changed the max action magnitude to .1 from .05. We define the low level action space for this suite to use 3DOF end-effector control along with grasp control; we implement the primitives using this action space.

## D   Primitive Implementation Details

In this section, we provide specific implementation details regarding the primitives we use in our experiments. In particular, we use an end-effector pose controller as $C_k$ for all $k$. We compute the target state $s^*$ using the components of the robot state which correspond to the input arguments of the primitive, $s_{\text{args}}$. We compute $s^*$ using the formula $s^* = s_{\text{args}} + \text{args}$. The error metric is computed in a similar manner $e_k = s^* - s_{\text{args}}$ across primitives. Returning to the lifting primitive example in the main text, $s_{\text{args}}$ would be the z position of the end-effector, $s^*$ would be the target z position after lifting, and $e_k$ would be the difference between the target z position and the current z position of the end-effector. In Table 5 we provide additional details regarding each primitive including the search spaces, number of low-level actions and which environment it was used in. One primitive of note is go to pose (delta) which performs delta position control. Using this primitive alongside the grasp and release primitives corresponds closely to the raw action space for Metaworld and Robosuite, environment suites in which we do not use orientation control.

We tuned the low-level actions per environment suite, but one could alternatively design a tolerance threshold and loop until it is achieved to avoid any tuning. We chose a fixed horizon which runs significantly faster and any inaccuracies in the primitives are accounted for by the learned policy. Finally, we do not use every primitive in every domain, yet across all tasks within a domain we use the same library. In Metaworld, the raw action space does not allow for orientation control so we do not either. Enabling orientation control with primitives can, in certain cases, make the task easier, but we do not include the x-axis and y-axis rotation primitives for fair comparison. In Robosuite, the default action space has orientation control. We found orientation control was unnecessary in order to solve the lifting and door opening tasks when we disabled orientation control for raw actions and for primitives. As a result, in this work we report results without orientation control in Robosuite.

| Hyper Parameter | Value |
| --- | --- |
| Actor output distribution | Truncated Normal |
| Discount factor | 0.99 |
| $\lambda_{GAE}$ | 0.95 |
| actor and value function learning rates | 8e-5 |
| world model learning rate | 3e-4 |
| Imagination horizon | 15 |
| Entropy coefficient | 1e-4 |
| Predict discount | No |
| Target value function update period | 100 |
| reward loss scale | 2 |
| Model hidden size | 400 |
| Stochastic state size | 50 |
| Deterministic state size | 200 |
| Embedding size | 1024 |
| RSSM hidden size | 200 |
| Use GRU layer norm | Yes |
| Actor hidden layers | 4 |
| Value hidden layers | 3 |
| batch size | 50 |
| batch length | 50 |

Table 2: Dreamer hyper-parameters

# E   RL Implementation Details

Whenever possible, we use the original implementations of any method we compare against. We use standard implementations for each base RL algorithm except Dreamer, which we implement in PyTorch. We use the actor and model hyper-parameters from Dreamer-V2 [19] as we found it slightly improved the performance of Dreamer. For primitives, we made several hyper-parameter changes to better tailor Dreamer to hybrid discrete-continuous control. Specifically, instead of back-propagating the return through the dynamics, we use REINFORCE to train the actor in imagination. We additionally reduce the imagination trajectory length from 15 to 5 for the single task primitive experiments since the trajectory length is limited to 5 in any case. With the short trajectory lengths in RAPS, imagination often goes beyond the end of the episode, so we use a discount predictor to downweight imagined states beyond the end of the episode. Finally since we cannot sample batch lengths of 50 from trajectories of length 5 or 15, we instead sample the full primitive trajectory and change the batch size to be $\frac{2500}{H}$, the primitive horizon. This results in an effective batch size of 2500, which is equal to the Dreamer batch size of 50 with a batch length of 50.

In the case of SAC, we use the implementation of SAC [29] but without data augmentation, which amounts to using their specific pixel encoder which we found to perform well. Finally for PPO, we use the following implementation: Kostrikov [28]. See Tables 2, 3, 4 for the hyper-parameters used for each algorithm respectively. We use the same algorithm hyper-parameters across all the baselines. For primitives, we modify the discount factor in all experiments to $1 - \frac{1}{H}$, in which $H$ is the primitive horizon. This encourages the agent to highly value near term rewards with short horizons. For single task experiments, we use a horizon of 5, taking 5 primitive actions in one episode, with a discount of 0.8. For the hierarchical control experiments we use a horizon of 15 and a corresponding discount of .93. In practice, this method of computing the discount factor improves the performance and stability of RAPS.

For each baseline we use the original implementation when possible as an underlying action space for each RL algorithm. For VICES, we take the impedance controller from the iros_19_vices branch and modify the environment action space to output the parameters for the controller. For PARROT, we use an unreleased version of the code provided by the original authors. For SPIRL, we use an improved version of the method which was released to the SPIRL code base recently. This version, SPIRL-CL, uses a closed loop decoder to map latents back to action trajectories which they find significantly improves performance on the Kitchen environment from state input. We use the authors' code for vision-based SPIRL-CL and still find that RAPS performs better.

| Hyper Parameter | Value |
|---|---|
| Discount factor | 0.99 |
| actor, critic, encoder learning rates | 2e-4 |
| alpha learning rate | 1e-4 |
| Target network update frequency | 2 |
| Polyak averaging constant | .01 |
| Frame stack | 4 |
| Image size | 64 |
| Random policy warm up steps | 2500 |
| batch size | 512 |

Table 3: SAC hyper-parameters

| Hyper Parameter | Value |
|---|---|
| Entropy coefficient | .01 |
| Value loss coefficient | 0.5 |
| Actor-value network learning rate | 3e-4 |
| Number of mini-batches per epoch | 10 |
| PPO clip parameter | 0.2 |
| Max gradient norm | 0.5 |
| $\lambda_{GAE}$ | 0.95 |
| Discount factor | 0.99 |
| Number of parallel environments | 12 |
| Frame stack | 4 |
| Image size | 84 |

Table 4: PPO hyper-parameters

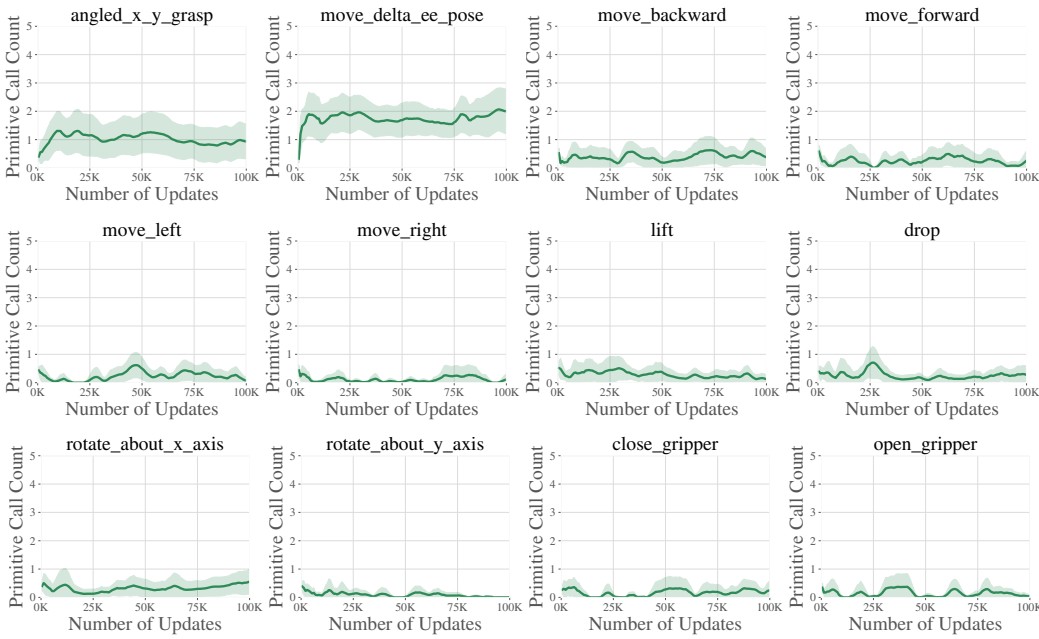

Figure 12: Primitive call counts for the evaluation policy averaged across all six kitchen tasks, plotted against number of training calls. In the beginning, each primitive is called at roughly the same frequency (uniformly at random), but over time the learned policies develop a preference for the dummy primitive and the angled xy grasp primitive, while still occasionally using the other primitives as necessary.

| Primitive Skill | Parameters | Action Space | # low-level actions | Environments |
|---|---|---|---|---|
| grasp | d | $[0,1]$ | 150-200 | Kitchen, Metaworld, Robosuite |
| release | d | $[-1,0]$ | 200-300 | Kitchen, Metaworld, Robosuite |
| lift | z | $[0, 1]$ | 40-300 | Kitchen, Metaworld, Robosuite |
| drop | z | $[-1, 0]$ | 40-300 | Kitchen, Metaworld, Robosuite |
| push | y | $[0, 1]$ | 40-300 | Kitchen, Metaworld, Robosuite |
| pull | y | $[-1, 0]$ | 40-300 | Kitchen, Metaworld, Robosuite |
| shift right | x | $[0, 1]$ | 40-300 | Kitchen, Metaworld, Robosuite |
| shift left | x | $[-1, 0]$ | 40-300 | Kitchen, Metaworld, Robosuite |
| go to pose (delta) | x,y,z | $[-1, 0]^3$ | 40-300 | Kitchen, Metaworld, Robosuite |
| x-axis twist | $\theta$ | $[-\pi, \pi]$ | 300 | Kitchen |
| y-axis twist | $\theta$ | $[-\pi, \pi]$ | 300 | Kitchen |
| angled forward grasp | $\theta$, x, y, d | $[-\pi, \pi], [-1, 0]^3$ | 1100 | Kitchen |
| top z grasp | z,d | $[-1, 0]^2$ | 140-250 | Robosuite |
| top grasp | x,y,z,d | $[-1, 0]^4$ | 1500 | Metaworld |

Table 5: Description of skill parameters, search spaces, low-level actions and environment usage.

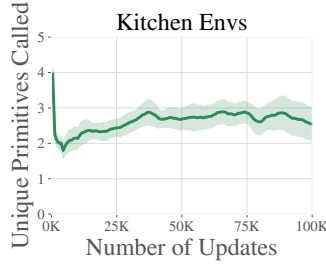

Figure 13: Number of unique primitives called by the evaluation policy averaged across all six Kitchen tasks, plotted against the number of training calls. Early on in training, the number of unique primitives called is four. With a path length of five this makes sense, on average it is calling unique primitives almost every time. At convergence, the number of unique primitives called is around 2.69. This suggests later on the policy learns to select certain primitives more often to optimally solve the task.

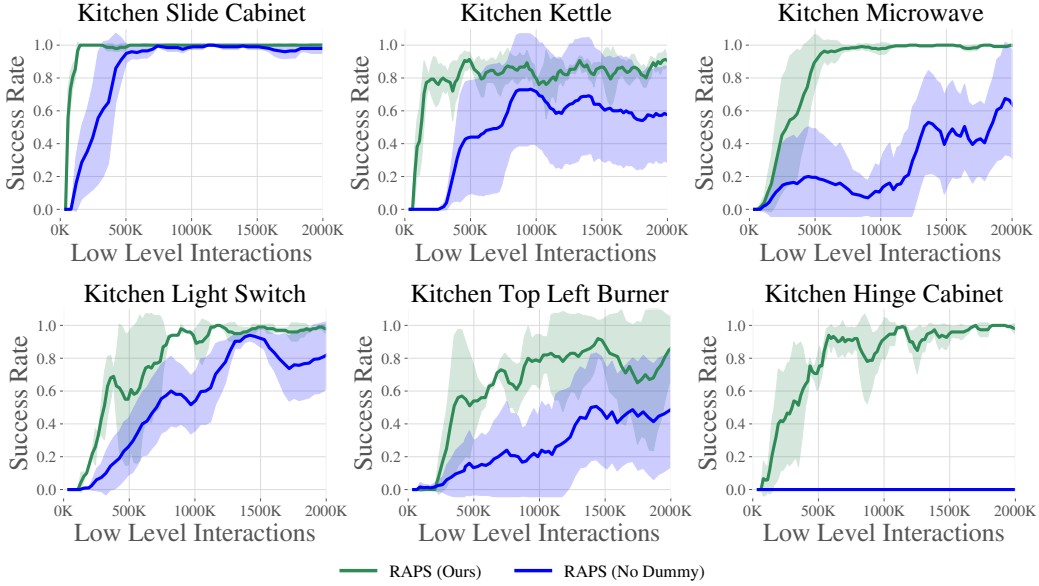

Figure 14: We run an ablation of RAPS in which we remove the dummy primitive, and we find that in general, this negatively impacts performance. Without the dummy primitive, RAPS is less stable and unable to solve the `hinge-cabinet` task.

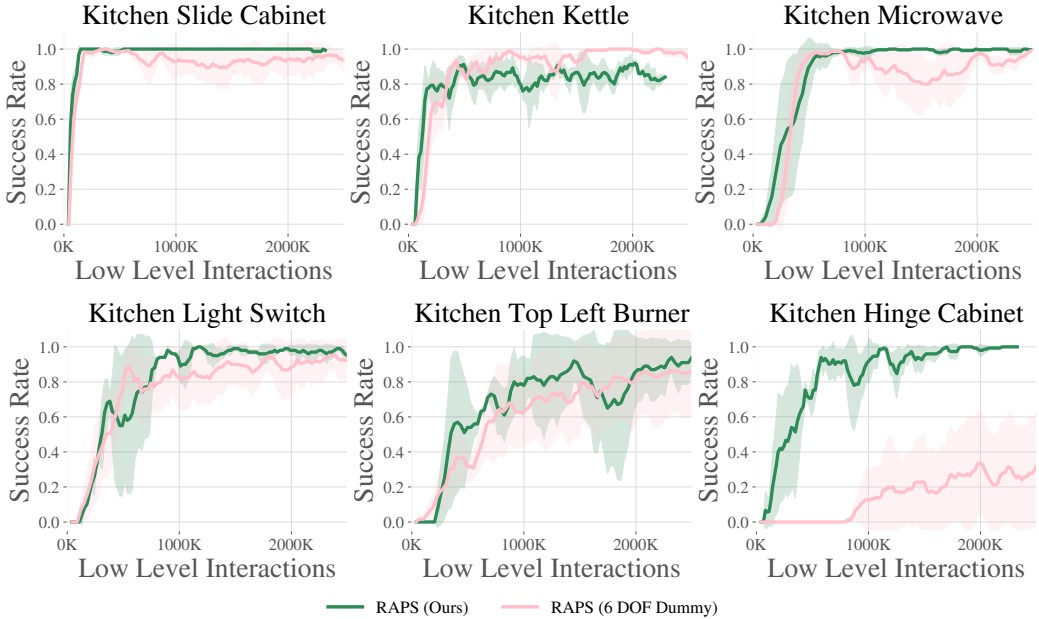

Figure 15: We plot the results of running RAPS with a 6DOF control dummy primitive, and find that in general, the performance is largely the same.

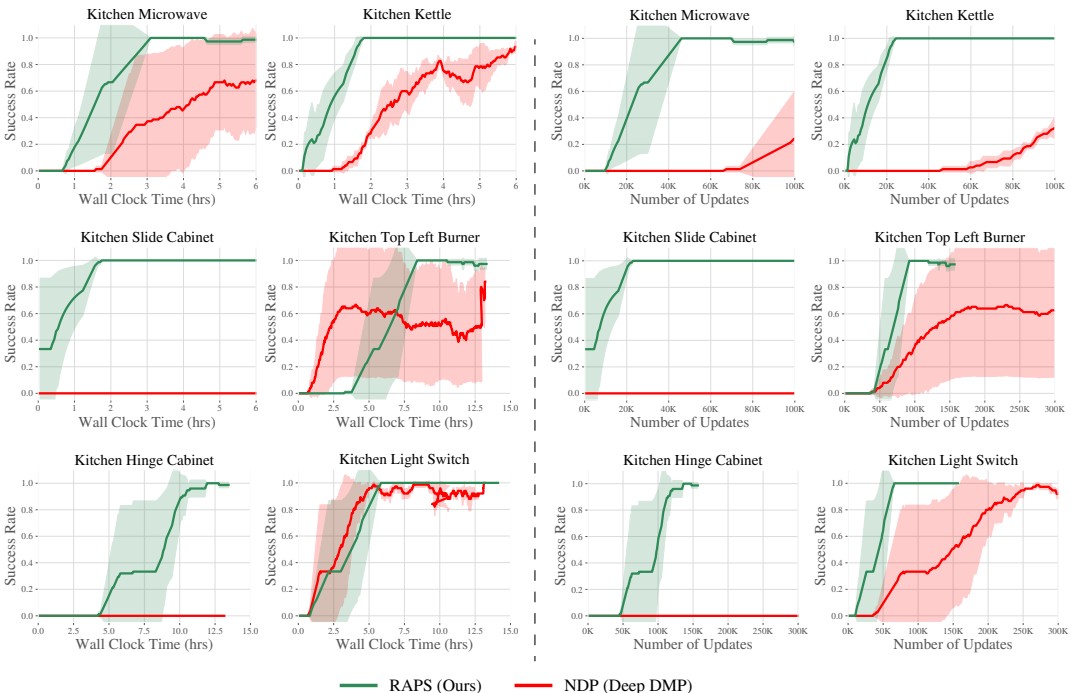

Figure 16: Comparison of RAPS against NDP, a deep DMP method for RL. RAPS dramatically outperforms NDP on nearly every task from visual input, both in terms of wall-clock time and number of training steps. This result demonstrates the increased capability of RAPS over DMP-based methods.

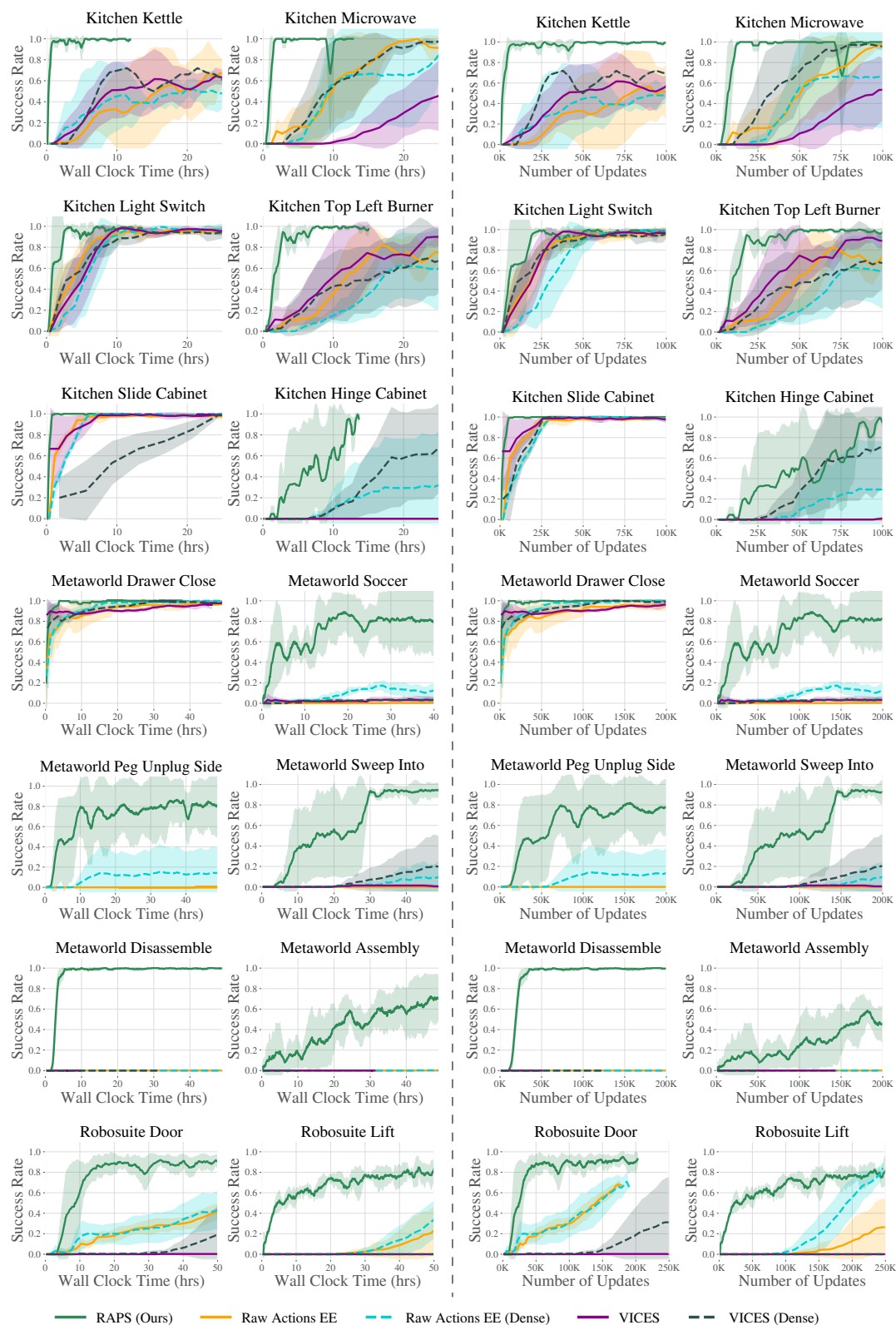

Figure 17: Full version of Figure 3 with excluded environments (`slide-cabinet` and `soccer-v2`) and plots against number of updates (right two columns). RAPS outperforms all baselines against number of updates as well.

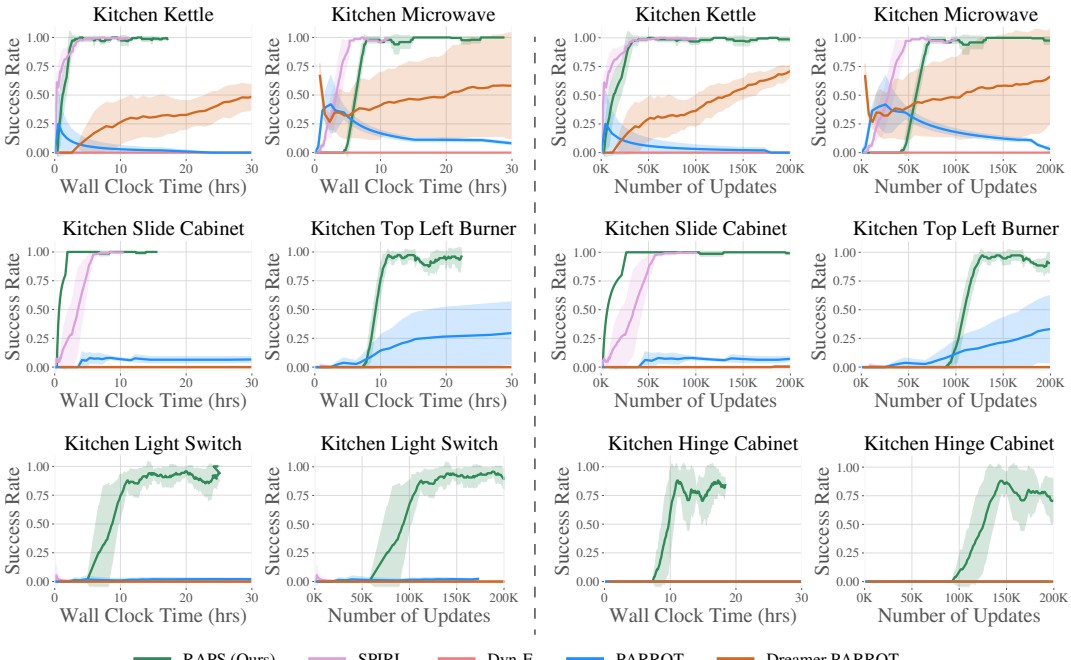

Figure 18: Full version of Figure 4 with plots against number of updates (right column) and excluded environments (`light-switch`). While SPIRL is competitive with RAPS on the easier tasks, it fails to make progress on the more challenging tasks.

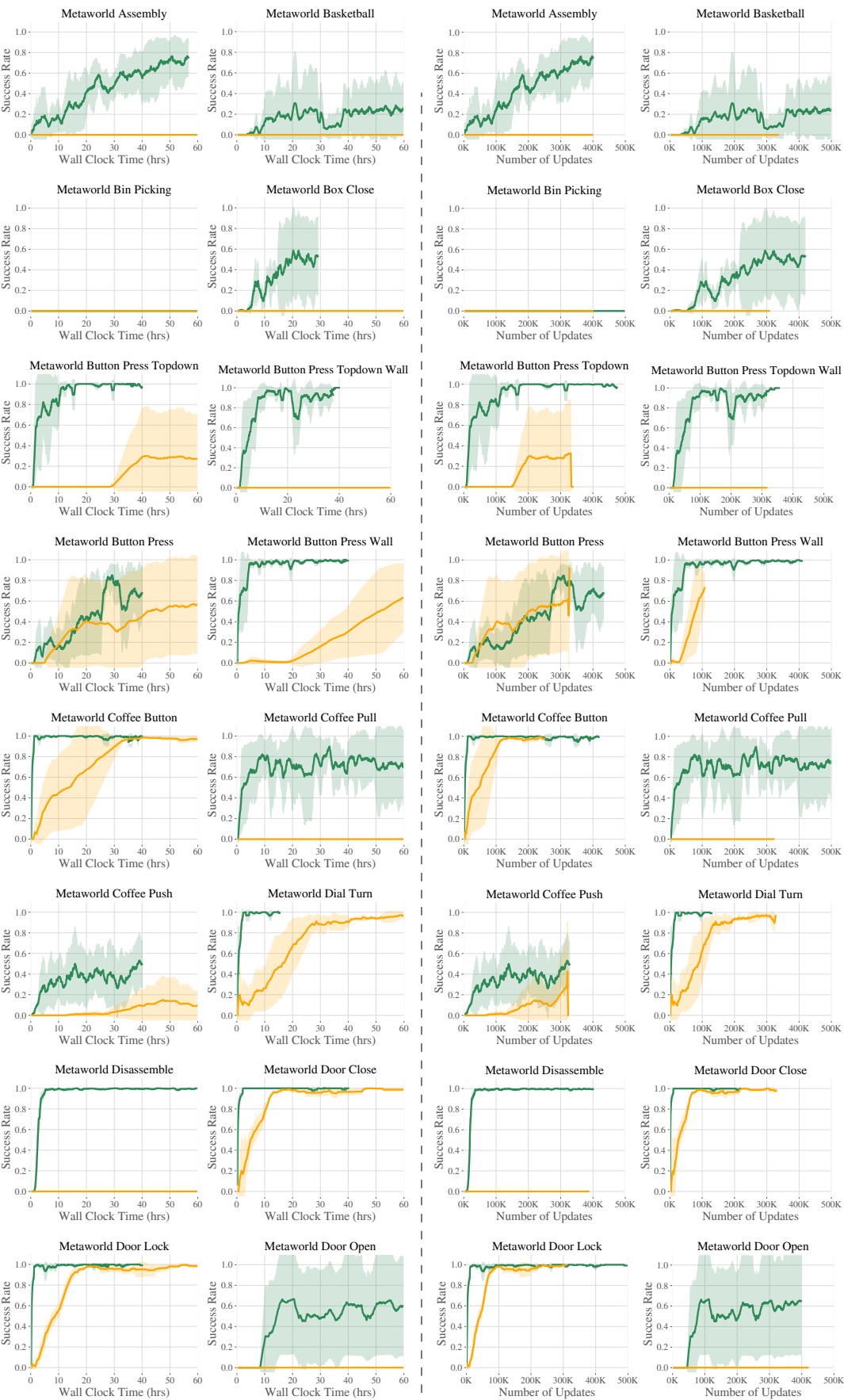

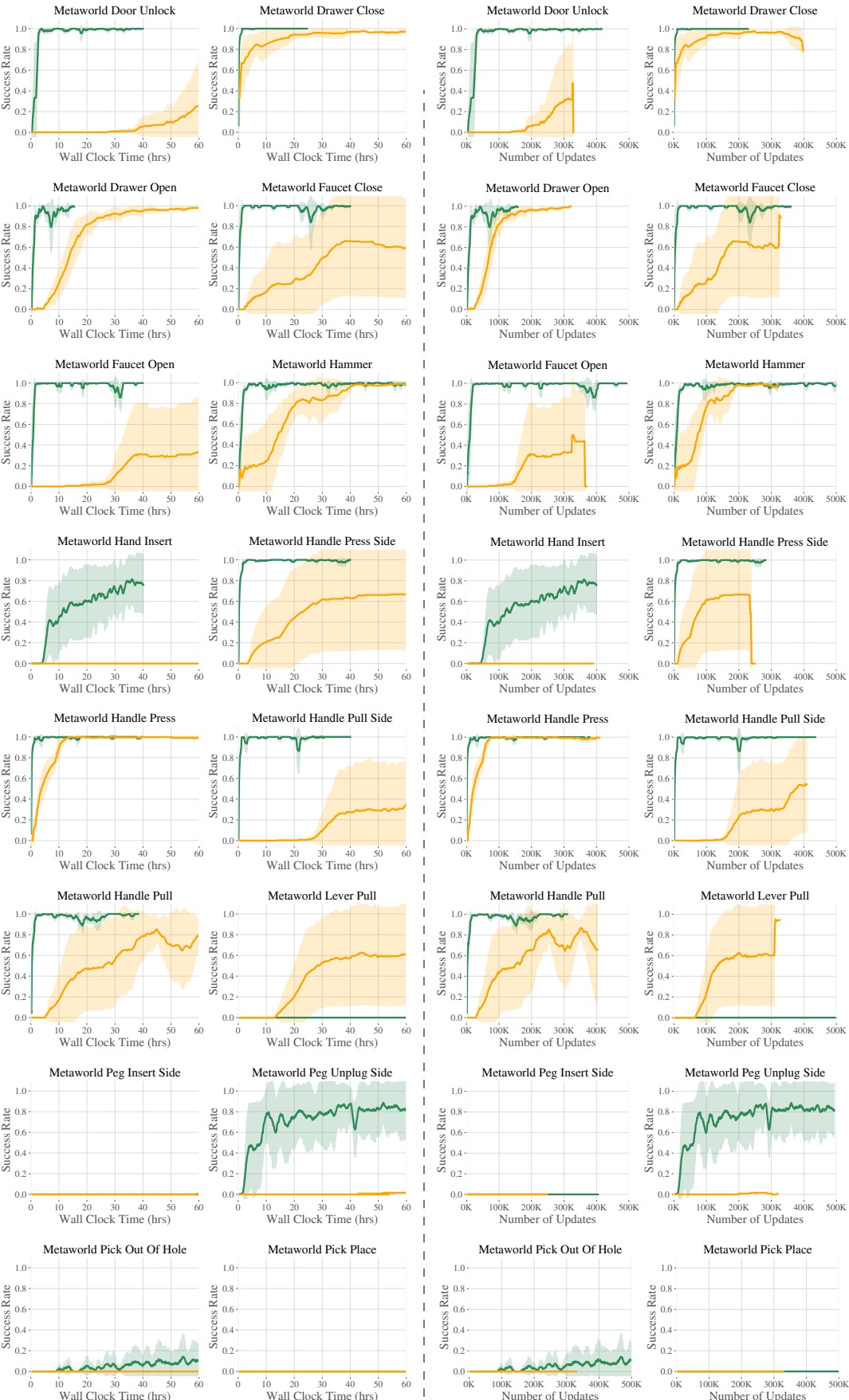

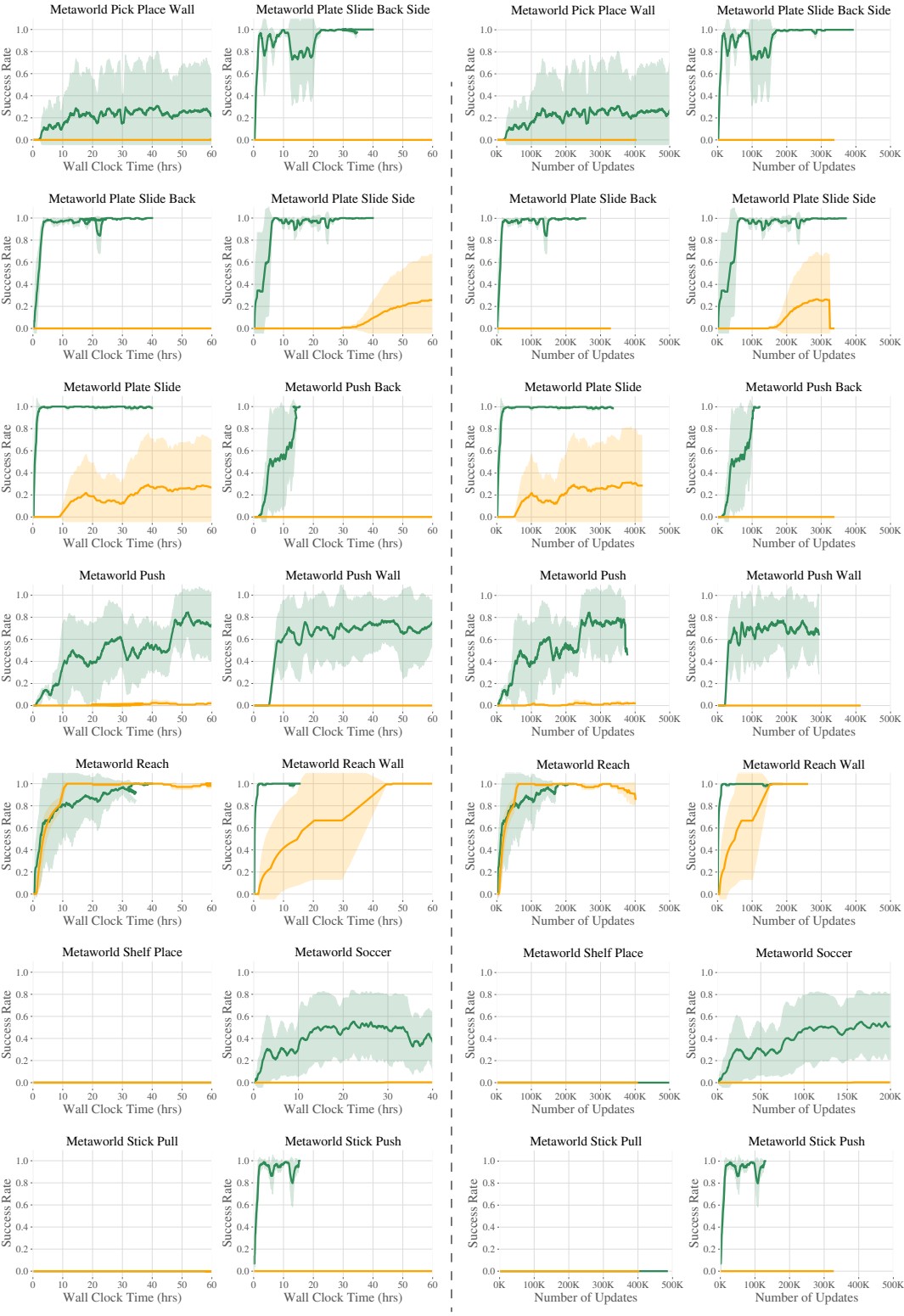

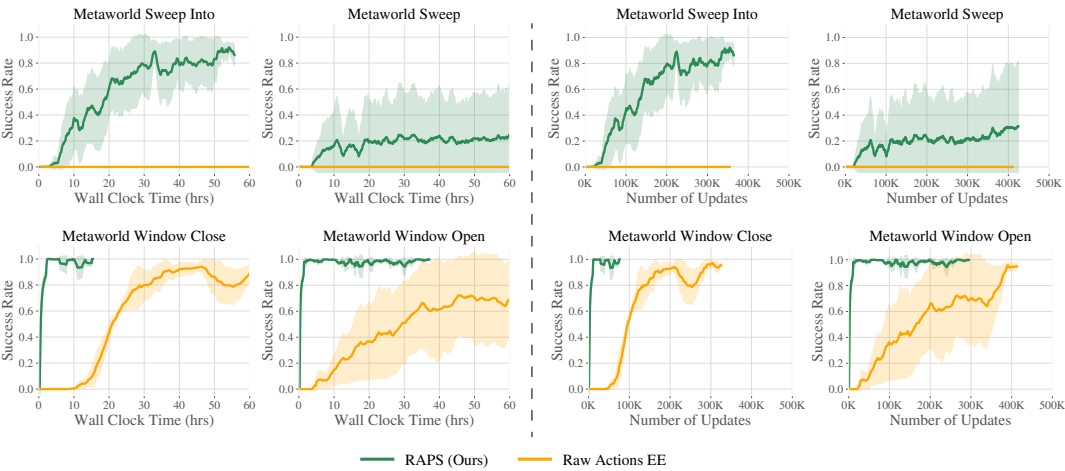

Figure 18: Comparison of RAPS against raw actions across all 50 Metaworld tasks from sparse rewards. RAPS is able to outright solve or make progress on up to **43 tasks** while Raw Actions struggles to make progress on most environments.