# OpenReview forum: "Accelerating Robotic Reinforcement Learning via Parameterized Action Primitives"
_NeurIPS.cc/2021/Conference — NeurIPS 2021 Poster_

### Official Review · Reviewer_mpsj · 2021-07-06

**Rating:** 6
**Confidence:** 4

**Summary:**

This paper proposes a novel action space for efficient learning of robotic manipulation tasks. In contrast to typical joint space and task space control, the proposed action space consists of a set of hand-designed agent-centric primitive skills (e.g. lift arm for $d$ cm, close grippers to $d$ cm, and rotate wrist for $\theta$) with continuous parameters (e.g. $d$ and $\theta$). Since the primitive skills are agent-centric (i.e. conditioned only on a robot state), the primitive skills can be used across diverse environments and tasks. With this appropriately designed action space, a high-level policy learns to choose which primitive skill and corresponding parameters to execute from visual inputs and sparse rewards.

The exhaustive experiments on three robotic manipulation benchmarks and three different RL algorithms demonstrate improved sample efficiency of the proposed action space. The proposed method outperforms baselines with joint space control, task space control, and learned skills from offline data or random interactions.

**Limitations And Societal Impact:**

Adequately addressed.

**Main Review:**

**Originality**: This paper proposes a novel action space that consists of multiple manually designed primitive skills, each of which is parameterized with continuous variables. This idea is somewhat similar to the idea of motion primitives but could be designed more general to different environments and tasks.

**Clarity**: The paper is clearly written and easy to follow. It would be great if more details about the primitive skills and how to derive this set of skills are discussed in the main paper.

**Quality**: The experiments are thorough and the proposed method successfully tackles many challenging manipulation tasks, which is impressive.

**Significance**: The primitive skills are tailored to the robots and tasks, which requires domain knowledge and human efforts for designing them. The paper claims that once the primitive skills are designed, they can be transferred to other tasks and environments, but it might be still limited to similar tasks and environments, which may require new set of primitive skills for new tasks and environments. However, the proposed method seems timely as this is a simpler alternative of widely used task space and joint space control, and provides significant improvements in exploration and sample efficiency.

The paper claims that the proposed action space is suited to learn from visual inputs and sparse rewards. Intuitively, this is true as the set of well-designed primitive skills can always provide meaningful transitions, so reduce the exploration burden. However, this is generally very challenging to learn a policy from high-dimensional visual inputs and sparse reward. Especially given that the action space is not smaller than the original action space (either joint torque control or end-effector control), training the high-level policy may require significant engineering efforts. Could you elaborate whether this was not the case or there were some important implementation or training details.

The proposed action space is tailored to the robots and tasks. The reviewer wonders how diverse set of tasks can be covered with the proposed action space, which set of tasks cannot be handled with the proposed action space (e.g. velocity, force sensitive tasks like catching a ball, pouring, etc), and how this action space is scalable, as more number of primitive skills and parameters would be required for complex tasks but it makes learning of the high-level policy difficult.

It is great to compare sample efficiency in terms of both "wall clock time" and "training steps". However, the comparisons on "wall clock time" could be highly dependent on implementation of each method and environment. Moreover, the "training steps" can be prone to the proposed method as the proposed method uses longer-horizon low-level skills compared to baselines. If we want to simulate real-world scenarios, the environmental steps (the number of low-level interactions) can be the best evaluation metric as real-world interaction is the most slow, dangerous, and costly component in real-world RL. It would be great to see the comparisons over the number of interactions.

**Time Spent Reviewing:**

4

---

> ### Author Response · Authors · 2021-08-10
> **Response to Reviewer mpsj**
>
> We thank the reviewer for their valuable feedback.
>
> > *“but it might be still limited to similar tasks and environments, which may require new set of primitive skills for new tasks and environments.”*
>
> - To a certain extent, this point is valid. Defining primitive skills over end-effector control limits the expressivity of our RAPS framework to tasks that can be solved with end-effector control. Furthermore, there could be skills within the end-effector control space that do not exist directly as a primitive or combination of primitives, such as pouring. To handle this to some extent, we include a dummy primitive that re-introduces the raw action space, albeit in a form compatible with the RAPS framework. We refer the reviewer to paragraph 3 in section 3.2 of the main paper for details about this primitive. We also include additional discussion in the reply to Review WUWx. Additionally, we demonstrate on a wide range of tasks in the main paper that RAPS is quite effective. Since the initial submission, we ran RAPS on the entire suite of 50 Meta-world tasks and found that RAPS can solve **43/50 tasks** within one day of training per task, whereas even with 3 days per task, the raw action space of end-effector control only enables the solution of up to 21 tasks: [https://drive.google.com/file/d/1jvKZyUEjhcbVGKoS-UvmNFsSuTBelQEX/view](https://drive.google.com/file/d/1jvKZyUEjhcbVGKoS-UvmNFsSuTBelQEX/view)
> We plan to include these results in an updated version of the paper.
>
> > *“If we want to simulate real-world scenarios, the environmental steps (the number of low-level interactions) can be the best evaluation metric as real-world interaction is the most slow, dangerous, and costly component in real-world RL. It would be great to see the comparisons over the number of interactions.”*
> - We note in the Evaluation Metrics paragraph under Section 4 that measuring with respect to low-level samples would make all higher level skills appear deceptively inefficient. This metric would fail to catch the fact that low-level action spaces require significantly more calls to the sensing systems and forward passes through convolutional networks. To experimentally verify this point, we ran a randomly initialized neural network policy with end-effector control and RAPS on a real robot (xArm 6: visualized here [https://drive.google.com/file/d/1gWpB7dWB9TRP54XSx9AX1Y-OdjKivtNy/view](https://drive.google.com/file/d/1gWpB7dWB9TRP54XSx9AX1Y-OdjKivtNy/view) ) and averaged the times of running ten trajectories. Each primitive ran 200 low-level actions with a path length of five high level actions, while the low-level path length was 500. Note that RAPS has double the number of low-level actions of the raw action space within a single trajectory. With raw actions, each episode took 16.49 seconds while with RAPS, each episode lasted an average of 0.51 seconds, a **32x speed up**. We will include these timing results in the discussion regarding evaluation metrics.
>
> > *“if more details about the primitive skills and how to derive this set of skills are discussed in the main paper.”*
> - Due to space considerations, we had included this discussion in Section C in the Appendix. To summarize, each skill is based in a feedback control loop with end-effector low level actions. The input arguments are used to define a target state to achieve and the primitive executes a loop to drive the error between the robot state and the target robot state to 0. Crucially, this loop does not depend on the state of the environment, only the state of the robot (proprioceptive state), enabling our method to scale directly to vision-based tasks. We additionally include a description of every primitive in Table 5 in the Appendix, with the input space, search space, domain used and more details.
>
> > *“training the high-level policy may require significant engineering efforts. Could you elaborate whether this was not the case or there were some important implementation or training details.”*
> - As described in Section 3.2, we made some changes to the policy architecture, similar in spirit to Chitnis et al. The high level policy is hybrid discrete-continuous, outputting a distribution over one-hot vectors over primitives and a distribution over all the arguments of all the primitives. We found that this simple change to the architecture as well as modifying the discount factor for RAPS to be 1-1/H (.8 for most experiments as H was generally 5) proved to be sufficient in achieving the results described. The aforementioned change to the discount factor is described in the Appendix. More importantly, for a discussion as to why the primitives are so effective even though in terms of size, the action space is actually significantly higher dimensional than for raw actions, we refer the reviewer to paragraph 3 in Section 3.2. In short, the decomposition between the **what** and the **how** enables significantly improved performance, as the higher level policy need only figure out what the task is as well as a high level plan to achieve it.
>
> > *“how this action space is scalable, as more number of primitive skills and parameters would be required for complex tasks but it makes learning of the high-level policy difficult.”*
>
> - As formulated in the paper at the moment, there is no explicit handling of scaling to a large number of primitives, this remains an active area of research. One promising solution, as proposed by SPIRL (CORL 2020), would be to learn a prior over skills, perhaps informed by an offline dataset. Since our skills are parameterized by arguments, they exhibit significant variation in behavior while remaining small in number. We use only up to 12 primitives in a single experiment. As a result, even with additional primitives added, we believe it is unlikely that the number of behaviors would scale well beyond what the current framework can handle, provided that each skill is parametric.

---

> > ### Comment · Reviewer_mpsj · 2021-08-14
> > **Response to the authors**
> >
> > Thank you for your detailed responses.
> >
> > > Since the initial submission, we ran RAPS on the entire suite of 50 Meta-world tasks and found that RAPS can solve 43/50 tasks within one day of training per task, whereas even with 3 days per task, the raw action space of end-effector control only enables the solution of up to 21 tasks: https://drive.google.com/file/d/1jvKZyUEjhcbVGKoS-UvmNFsSuTBelQEX/view
> >
> > I appreciate that the authors reported the additional experimental results on MetaWorld and it is great to see that the proposed method works across a diverse set of tasks in MetaWorld. However, as most tasks in MetaWorld are about short-horizon reaching, pushing, picking, and placing, the doubt on the limitation of the proposed action space still exists. But, this might be okay as the proposed method is a simple yet effective way of tackling many manipulation tasks in a timely manner.
> >
> >
> > > “if more details about the primitive skills and how to derive this set of skills are discussed in the main paper.”
> >
> > I would like to ask including a brief explanation about the primitive skills in the main paper, not all the details in the appendix. Since the set of primitive skills used in this paper is one of the major contributions rather than the framework itself, explaining how the authors design the action space would be valuable to discuss in the paper.
> >
> >
> > > To handle this to some extent, we include a dummy primitive that re-introduces the raw action space, albeit in a form compatible with the RAPS framework.
> >
> > Can you provide more details about training the whole framework with the dummy primitive? Section 3.2 does not include any details about how the dummy primitive works and is trained. Also, having such a dummy primitive in a hierarchical framework is known to be tricky to train and requires balancing the execution between this dummy primitive and other options. Since the paper shows successful training with the dummy primitive, I wonder what makes the proposed method work this well with the dummy primitive as well as how much impact this dummy primitive has (e.g. w/ and w/o dummy primitive; and how many times the dummy primitive is used in each episode).
> >
> > As I went through some of the implementation details, I'm a bit confused about the action spaces of Raw Action and VICES baselines. In L263, the paper describes that the Raw Action and VICES use 6DOF end-effector control instead of joint velocity control. Is this true for all experiments in the paper (just want to double-check)? Also, is the dummy primitive also working with the 6DOF end-effector control?
> >
> >
> > > We note in the Evaluation Metrics paragraph under Section 4 that measuring with respect to low-level samples would make all higher level skills appear deceptively inefficient. This metric would fail to catch the fact that low-level action spaces require significantly more calls to the sensing systems and forward passes through convolutional networks.
> >
> > I still do not agree with the authors that the number of low-level interactions is not a good measure. Based on your current implementation this might be true. However, a single pass of a neural network can become much faster with better computation units and optimization in a near future (based on the current trend), while the risk and cost of running robots have been decreasing very slowly. Thus, I think comparing the number of low-level interactions is very important especially in robotics. I would like to make sure this comparison is included in the main paper; otherwise, I probably lower down my rating.

---

> > > ### Author Response · Authors · 2021-08-16
> > > **Response to follow up from Reviewer mpsj**
> > >
> > > We thank the reviewer for their detailed response. We have run the additional experiments you requested and plotted them here: https://drive.google.com/file/d/1jP6Xuryw5Swap1kuUNeD7vcT9ER8s9RX/view
> > >
> > > In the responses below, we address specific points in detail.
> > >
> > > > *”Thus, I think comparing the number of low-level interactions is very important, especially in robotics. I would like to make sure this comparison is included in the main paper”*
> > > - As per the reviewer’s request, we will include these plots in the paper. Sorry for the delay in our response, we were running the experiments by logging low-level steps. Note just as a reminder, our RAPS higher-level policy takes significantly fewer policy gradient updates (and hence, takes much less wall clock time) than the baselines for the same number of low-level interactions, hence the comparison of low-level interactions is a little unfair but we will include it in the paper nevertheless. Results are here: https://drive.google.com/file/d/1dTq0MzFP-aSL19rSdbHwsP2vSPIIfF__/view. As shown in the plots, RAPS still improves in terms of performance over the baselines, though the efficiency gain with respect to low-level actions is not as dramatic as with respect to wall-clock time or the number of policy updates. We will add more plots in the paper as they are currently running since the wall clock time of the baselines is slow and takes many days for just one run (although it may not be apparent from the number of low-level interactions).
> > >
> > > > *”Can you provide more details about training the whole framework with the dummy primitive?”*
> > > - To clarify, the dummy primitive is not trained. It is a simple placeholder and proxy for low-level actions to provide enough freedom to the high-level policy to model parts of the trajectories that can not modeled by the designed RAPS primitives.
> > > The dummy primitive directly takes in a delta position and then tries to achieve it by taking a fixed number of steps. This is similar to the raw action space which also operates as end-effector control but the difference is in frequency. The dummy primitive only returns to the high-level policy after a fixed number of steps while raw-actions return observations after each step.
> > >
> > > > *”I wonder what makes the proposed method work this well with the dummy primitive as well as how much impact this dummy primitive has (e.g. w/ and w/o dummy primitive; and how many times the dummy primitive is used in each episode).”*
> > > - As explained above, one of the core reasons the dummy primitive works so well is that it is doing low frequency end-effector control. The dummy primitive is far more accurate than the low-level action space, as it spends more than one step attempting to achieve the policy’s desired delta pose, at the cost of expressiveness. For a discussion of this cost, please see the reply to reviewer WUWx, point (1).
> > > - **How often is the dummy primitive used?**: The dummy primitive is actually one of the two most used primitives (also known as move delta ee pose), the other being angled xy grasp (also known as angled forward grasp in the appendix). The plots over how many times the dummy primitive is used per episode over the course of training is plotted here: https://drive.google.com/file/d/1T1YfEWzT7nV5aD4Q2iqiGAgMBigTyiDz/view , with a discussion of the results in the reply to reviewer WUWx.
> > > - **How useful is dummy primitive?**: We have additionally run an experiment without the dummy primitive in order to evaluate its impact, and it shows that the dummy primitive helps quite a lot as hand-designed primitives may not always be sufficient. Plots are linked here: https://drive.google.com/file/d/1-xMLm-0CjVn-Zack0jp7Ti1Wg7hmDz65/view
> > >
> > > > *”In L263, the paper describes that the Raw Action and VICES use 6DOF end-effector control instead of joint velocity control. Is this true for all experiments in the paper (just want to double-check)? Also, is the dummy primitive also working with the 6DOF end-effector control?”*
> > > - The raw action space and VICES use 6DOF end-effector control with grasp control only in the Kitchen environments. In the Meta-world and Robosuite domains, the raw action space and VICES use 3DOF end-effector control (cartesian control only) along with grasping, as this action space is sufficient to solve every task. We will clarify this line in the paper.
> > > - The dummy primitive uses 3DOF control in the experiments in the paper, but it could just as easily do 6DOF control if desired. In fact, we tested this and RAPS still works with 6DOF dummy primitives: https://drive.google.com/file/d/1-xMLm-0CjVn-Zack0jp7Ti1Wg7hmDz65/view
> > >
> > > > *”I would like to ask for a brief explanation about the primitive skills in the main paper, not all the details in the appendix.”*
> > > - Yes, we get one extra page in the camera ready for the main paper, so we will shift part of the detailed description of the primitive skills from the appendix to the main paper.

---

> > > > ### Comment · Reviewer_mpsj · 2021-08-19
> > > > **Response to authors**
> > > >
> > > > Thank you for the clarification and additional experiments. It addresses all my concerns and I have no more questions. I think the paper will be much more convincing with new results and anlaysis, and the clarification.

---

### Official Review · Reviewer_cseK · 2021-07-09

**Rating:** 6
**Confidence:** 5

**Summary:**

Low-level actions, such as torques or joint-velocities, provide a hard-exploration challenge for applying reinforcement learning (RL) to robotic systems. For example, it might be impossible to reach and grasp an object solely due to random torques or joint velocities. To overcome this difficulty, one can define high-level actions to enable a meaningful exploration. In this paper, the authors introduce parametrized action primitives, which encode a robotic movement and use those actions for learning with RL.
The parametrized actions are just manually encoded primitives, such as moving the arm up, reaching a position $x$ and grasp, and so on.
Therefore, the system proposed by the authors requires expert knowledge of the system, the task, and robotics.
This simplification of the task results, according to the paper, in a more efficient RL.

**Ethical Concerns:**

I do not recognize any ethical concern with this work.

**Limitations And Societal Impact:**

The authors discussed the limitations of their work.
The societal impact was not addressed, but, in all fairness, I believe was not necessary.

**Main Review:**

Albeit I strongly agree with the authors that RL cannot be performed on "low-level" action (especially if one wants to learn directly on a real system), I struggle to find any novelty or contribution.

The proposed parameterized actions are - in my understanding - just mere trajectory planners. I do not find any novelty in their introduction nor their use in RL. It is not surprising that the proposed framework is more efficient than baselines since the framework requires much more human effort. The authors argue that these action primitives are easy to implement (perhaps easy for a person who can program and understand a robotic system) and transferable between robots.
Naturally, the authors claim that this framework works with different RL techniques, such as model-based, model-free. Again, this is not surprising since what the authors are doing is to frame a different MDP where instead of low-level actions, we have high-level ones.
I wonder why the authors did not choose perhaps a smarter parametrization of robotic movements. The literature about movement primitives is abundant. Dynamic movement primitives (DMPs) [1] or probabilistic movement primitives (ProMPs) [2] are good examples of parametric movements that can be used along with RL. Traditionally, movement primitives can be automatically extracted by existing data using simple regression and segmentation techniques when needed.

In short, the authors propose just a higher-level action MDP, where any RL technique will benefit the abstraction. The method is not novel, does not give any new insights about RL or robotics. In my opinion, the proposed solution completely lacks novelty. Movement primitives have already been used in the context of reinforcement learning [3, 4].
I would recognize some novelty if the authors had proposed a novel technique for extracting movement primitives, or a new reinforcement learning technique specifically designed to use movement primitives.
In my perspective, none of the two has been done.

-------------- UPDATE ---------------
After the discussion with the authors, I am willing to give more credit to their work.


[1] Ijspeert, Auke Jan, et al. "Dynamical movement primitives: learning attractor models for motor behaviors." Neural computation 25.2 (2013): 328-373.
[2] Paraschos, Alexandros, et al. "Probabilistic movement primitives." Advances in neural information processing systems (2013).
[3] Stulp, Freek, and Stefan Schaal. "Hierarchical reinforcement learning with movement primitives." 2011 11th IEEE-RAS International Conference on Humanoid Robots. IEEE, 2011.
[4] Li, Zhijun, et al. "Reinforcement learning of manipulation and grasping using dynamical movement primitives for a humanoid-like mobile manipulator." IEEE/ASME Transactions on Mechatronics 23.1 (2017): 121-131.

**Time Spent Reviewing:**

4

---

> ### Author Response · Authors · 2021-08-10
> **Response to Reviewer cseK**
>
> Thank you for the review, we discuss your concerns below:
>
> > *“The method is not novel, does not give any new insights about RL or robotics.”*
>
> - While parametric primitives have had a big impact in robotics, this methodology is still not prevalent in Deep RL. A large majority of the work on Deep RL either still uses raw actions or attempts to discover such primitives via demonstration or reinforcement, such examples include: Option-Critic (AAAI 2017), HIRO (NeurIPS 2018), DIAYN (ICLR 2019), DADS (ICLR 2020), SPIRL (CoRL 2020), PARROT (ICLR 2020), Discovering Motor Programs by Recomposing Demonstrations (ICLR 2020), DYNE (ICLR 2020), Learning Robot Skills with Temporal Variational Inference (ICML 2021), OPAL (ICLR 2021), Skid-RAW (ICRA 2021) and many more.
>
> - The contribution of this work is to show that a simple yet intuitive method outperforms the top performing methods among this list by a significant margin across diverse setups commonly considered in the community. Furthermore, RAPS is able to solve many tasks that were unsolvable before, even with demonstrations. We also demonstrate that RAPS scales effectively to multi-step hierarchical control tasks, a highly challenging domain in both robotics and RL. Finally, RAPS is able to achieve this performance using **high-dimensional image input and just sparse terminal rewards.**
>
>
>
> > *“I wonder why the authors did not choose perhaps a smarter parametrization of robotic movements."*
>
> - Thank you for pointing this out. We describe the tradeoffs of using DMPs in our related works section: they are usually difficult to scale to high-dimensional image input and often rely on task specific demonstrations. In our work, we aim to avoid the usage of demonstrations, instead opting to pay a small one-time cost in designing primitives with no further human involvement in the robot learning process. We argue this methodology is more scalable as it is possible to test a primitive skill in isolation easily and then deploy it across a wide variety of robotic platforms, environments and tasks.
>
> - However, as per your suggestion, we compared our method RAPS with the latest state-of-the-art work that incorporates DMPs with deep RL: Neural Dynamic Policies (NeurIPS 2020, RSS 2021). Results are here: [https://drive.google.com/file/d/1Ws4XE8vvMjxQmdqtEZnsbb7E1K9V-OMh/view](https://drive.google.com/file/d/1Ws4XE8vvMjxQmdqtEZnsbb7E1K9V-OMh/view?usp=sharing) Across nearly every task in the Kitchen suite, RAPS outperforms NDP just as it outperforms all prior skill learning methods as well. We will include this baseline in the main paper.
>
> > *“Therefore, the system proposed by the authors requires expert knowledge of the system, the task, and robotics.”*
>
> - We respectfully disagree. At best, our method requires an end-effector controller implementation which most manipulators come with out of the box. We provide the code for the primitives so any new user would simply have to plug in the end-effector controller into it in order to directly start using RAPS. Additionally, our primitives are task agnostic in that they are general movements that are useful in a large variety of settings. We demonstrate this effectiveness across 16 different manipulation tasks in the original paper.
>
> - Upon suggestion from R3 # mpsj, we additionally ran an experiment after submission that demonstrates that RAPS scales to **43 out of 50** of the Meta-world manipulation tasks: [https://drive.google.com/file/d/1jvKZyUEjhcbVGKoS-UvmNFsSuTBelQEX/view](https://drive.google.com/file/d/1jvKZyUEjhcbVGKoS-UvmNFsSuTBelQEX/view)
>
>
>
> > *“It is not surprising that the proposed framework is more efficient than baselines since the framework requires much more human effort.”*
> - As mentioned above, the proposed framework does not involve much human effort or robotics knowledge. RAPS requires an end-effector controller implementation which most manipulators come with out of the box. Additionally, we believe that the result that such a simple approach outperforms almost all the recently proposed complex methods to discover primitives by a large margin across all setups is not an *unsurprising* conclusion, especially given that with parameterization, the action space becomes significantly higher dimensional than the raw actions.

---

> > ### Comment · Reviewer_cseK · 2021-08-11
> > **Response to the authors.**
> >
> > Thank you for your detailed answer. You are completely right in pointing out that a large portion of the literature proposes the direct application of deep RL to raw actions. However, the usage of movement primitives in RL is not novel:
> >
> >
> > Stulp, Freek, and Stefan Schaal. "Hierarchical reinforcement learning with movement primitives." 2011 11th IEEE-RAS International Conference on Humanoid Robots. IEEE, 2011.
> >
> > K Ploeger, M Lutter, J Peters. "High Acceleration Reinforcement Learning for Real-World Juggling with Binary Rewards"
> > Conference on Robot Learning (CoRL).  (here the movement primitive is specified on a set of via-points)
> >
> > S., Tosatto, C., Georgia, J., Peters. "Contextual Latent-Movements Off-Policy Optimization for Robotic Manipulation Skills", ICRA 2021
> >
> > Li, Zhijun, et al. "Reinforcement learning of manipulation and grasping using dynamical movement primitives for a humanoid-like mobile manipulator." IEEE/ASME Transactions on Mechatronics 23.1 (2017): 121-131.
> >
> > The options framework introduced in 1999, foresees the possibility of working at two different levels of  action abstraction
> >
> > Sutton, Richard S., Doina Precup, and Satinder Singh. "Between MDPs and semi-MDPs: A framework for temporal abstraction in reinforcement learning." Artificial intelligence 112.1-2 (1999): 181-211.
> >
> > You say that the implementation of the primitive is simple as one can work just in the task space (commanding the 3D position of the end effector).
> >
> > While that is undoubtedly true, To understand how to approach an object (for example to open a drawer), or how to pick it up is non-trivial. To start, acquiring the 3d position of objects is non-trivial. Understanding how to approach complex objects is nontrivial too. There is an entire field that studies how to automotive grasping, which is, in general, a very complex problem. I don't buy that the manual encoding of movement primitives is simple.
> >
> > I agree that working in the task space simplifies the problem, but the movement primitive still remains complex to be implemented.
> > A also disagree that the movement primitives are completely transferable from one robot to the other. In presence of obstacles, the shape of the robot matters, and for the same end-effector trajectories, different configurations of the joint-space might be needed depending on the robot used.
> >
> > In short, while I still like the line and the direction of the work, I think that its novelty is rather limited.
> > What has been done, in short, is to specify an abstract version of the MDP by inserting smarter high-level actions (that require human and domain knowledge to be implemented).

---

> > > ### Author Response · Authors · 2021-08-12
> > > **Response to followup from Reviewer cseK**
> > >
> > > Thank you for your prompt and detailed response.
> > >
> > > > *"In presence of obstacles, the shape of the robot matters, and for the same end-effector trajectories, different configurations of the joint-space might be needed depending on the robot used."*
> > > > *"To understand how to approach an object (for example to open a drawer), or how to pick it up is non-trivial. To start, acquiring the 3d position of objects is non-trivial. Understanding how to approach complex objects is nontrivial too."*
> > > - We believe there might be a misunderstanding here about the kind of primitives we employ in our paper. Our RAPS primitives do not have any knowledge of the environment or the objects. The primitives themselves are *task-agnostic* and parameterized by arguments that are predicted by the policy. **Hence, the onus of behaving correctly around obstacles falls to the higher level policy which is learned by Deep RL and not the primitives themselves.** For instance, a reaching primitive cannot directly avoid an obstacle, hence the high-level policy must decide to stitch multiple of these primitives by predicting appropriate waypoints around the obstacle. Our primitives are just inverse kinematics (IK) controllers -- we hope this clarifies your concern!
> > > - Additionally, if there are behaviors that are not possible with the current primitives, we include a dummy primitive that can learn to adapt to the necessary motion, however, we believe there are better solutions possible that future work may investigate.
> > > - We empirically demonstrate that RAPS fully solved **43/50 manipulation tasks** from the Meta-world suite. In these tasks, RAPS is able to learn how to approach objects, acquire 3d positions and grasp complex objects -- even though the low-level primitives cannot on their own. Link: [https://drive.google.com/file/d/1jvKZyUEjhcbVGKoS-UvmNFsSuTBelQEX/view] (https://drive.google.com/file/d/1jvKZyUEjhcbVGKoS-UvmNFsSuTBelQEX/view)
> > >
> > > > *"I don't buy that the manual encoding of movement primitives is simple."*
> > > - In the previous point, we described why our primitives are simple to design and execute on a robot because they do not require specialized knowledge of grasp points or object approaches. We kindly request the reviewer to take a look at Section C in the supplementary for a detailed discussion of how our primitives are implemented.
> > >
> > > > *“also disagree that the movement primitives are completely transferable from one robot to the other.”*
> > > - We demonstrate empirically that transfer is possible in certain cases. In the original submission, we re-use the same primitive implementations across two different robots and three different simulators while still out-performing prior work.
> > > Additionally, in our response to reviewer 1 # WUWx, we find that the higher level policy can transfer 100% between morphologically different robots, such as the xArm 7 and xArm 6.
> > >
> > > > *“the usage of movement primitives in RL is not novel:...”*
> > > - To clarify the reviewers’ concern, we noted in paragraph 1 of our related works section that dynamic movement primitives (DMP) have been used in RL and this paper does not claim that the usage of primitives in RL is novel. That being said, the specific instantiation we describe in this work is *different* from DMPs and other movement primitive work as it is **task agnostic**, is parameterized by arguments that ensure behavior variation in each primitive, and includes a *dummy primitive* to reintroduce the raw action space.
> > > Additionally, in our earlier response as per the reviewer’s suggestion, we compared against the most recent state-of-the-art DMP+Deep RL method (NDP) and showed that RAPS outperforms it by a significant margin across a wide range of tasks. We would like to bring the reviewer’s attention to those results: [https://drive.google.com/file/d/1Ws4XE8vvMjxQmdqtEZnsbb7E1K9V-OMh/view] (https://drive.google.com/file/d/1Ws4XE8vvMjxQmdqtEZnsbb7E1K9V-OMh/view)
> > >
> > > > *“The options framework introduced in 1999, foresees the possibility of working at two different levels of action abstraction”*
> > > - We have discussed this connection in paragraph 2 of the related works section. In particular, the primitives we describe in this work can be viewed under the lens of the options framework.
> > >
> > > **## Main contribution of this work**
> > >
> > > Finally, we kindly urge the reviewer to take a step back and look at the bigger picture. While primitives have been studied in robotics, they are yet to be adopted in mainstream *Deep* RL where many papers are written regularly on how to discover primitives (see list in our earlier response). We show that a simple implementation of parameterized primitives combined with Deep RL significantly outperforms almost all of the state-of-the-art methods proposed previously across 60+ environments and across setups ranging from model-free to model-based, from sparse reward to no reward. This thorough empirical evidence is the main contribution of this work, as acknowledged by other reviewers as well, and we believe that this framework will become a strong baseline to beat for future hierarchical deep RL methods.

---

> > > > ### Comment · Reviewer_cseK · 2021-08-13
> > > > **Followup**
> > > >
> > > > I do understand that the action primitive is a higher-level action. Therefore it does not depend on the state of the environment (instead, the policy selects action primitives based on the state of the environment).
> > > >
> > > > What I am arguing is the following. Even an IK controller does have different effects on different robots. If I select command my end-effector to reach a position (x, y, z), the joint trajectory will depend on the kinematic structure of the robot. The resulting movement will be different from robot to robot, and the success of the primitive will also depend on the robot. So, while I am conscious that I can define similar controllers to different robots, the effect will be substantially different.
> > > > Primitives like grasping will also depend on the kind of end effector, e.g., gripper, dextrous hand,...
> > > >
> > > > I believe that a different set of action primitives must be designed for a quadruped robot, a classical robot manipulator, a humanoid, a table-tennis setup, ...
> > > >
> > > > _However, I do not want to argue on the transferability of action-primitives, which is not the main point of your work, I believe._
> > > >
> > > > I looked at Section C in the appendix, and I think I understood what an action primitive is.
> > > >
> > > > As you know, DMPs are not the only framework of motion primitive existing. ProMPs do not necessarily depend on a state. One can design a policy that selects the parameter of the ProMPs and execute the ProMPs similarly as you are doing in your work. I do not see a large difference between the two approaches.
> > > >
> > > > I can recognize the benefit of having your action primitives. However, I keep arguing that you still need a (minimal, if you want) human intervention to identify the action primitives.
> > > >
> > > > In my perspective, what you are doing, as I wrote before, is to formulate an MDP with a more convenient action space. I am not surprised that your algorithm exhibits higher performance.
> > > >
> > > > Your setup works well in manipulation tasks, as you clearly show in the paper. I think that your action primitive framework could have problems in dynamic tasks like tennis-table, walking, or in general tasks where fast adaptability is required. I know that you identified a dummy-primitive; however, the possibility of selecting many options might actually result in a lover performance when the dummy primitive is the action that one needs most of the time.
> > > >
> > > > I see that you recognize that your paper is a special case of the option framework. And that is exactly why I argue that your contribution is limited. The concept of temporal abstraction is well established in the literature. Your paper is just proposing your action primitives as options in the context of robotics.
> > > >
> > > > I started my review by writing that I fully agree that acting directly on the torque space hinders the applicability of deep reinforcement learning to robotics.
> > > >
> > > > I fully acknowledge that your paper is well written, clear, and has a strong empirical section. I cannot deny that the contribution feels limited.
> > > >
> > > > Nevertheless, I will adjust my review to give your work more credit.

---

> > > > > ### Author Response · Authors · 2021-08-14
> > > > > **Response to Followup**
> > > > >
> > > > > We thank the reviewer for their careful consideration and follow-up discussions. As per the reviewers comments, we will incorporate this discussion of different kinds of robots and dynamic tasks into the paper. We will also expand our discussion of movement primitives.
> > > > >
> > > > > Finally, we really appreciate the reviewer’s prompt follow ups during the rebuttal phase and thank the reviewer for increasing their final score.

---

### Official Review · Reviewer_WUWx · 2021-07-17

**Rating:** 7
**Confidence:** 3

**Summary:**

The authors used a hierarchical setup to learn complex manipulation tasks. Instead of using raw joint controls, the the action space is composed of a selection of primitive actions including lifting, pushing, and dummy primitive action ,etc. Then at high level, a learned policy decides to choose which motion primitives and outputs the corresponding parameters. The authors demonstrated that using their method, RAPS, the robot arm can solve many benchmark manipulation tasks within shorter wall clock time, significantly outperform a few STOA baselines.

**Main Review:**

The main strength of this paper is that, using the RAPS architectures, many challenging tasks can be solved effectively in sim after a few hours of training. By constraining the action outputs to the motion primitive space, the framework is also insensitive to the specific learning algorithms being used. The paper is generally well written, though a few details warrant more discussion.  The following are may comments to address:

(1) The dummy primitive. The authors said "As the dummy primitive operates on the high level horizon for Hk steps when called. Therefore, it cannot execute every trajectory that a lower level policy could. " This is a bit confusing. If the dummy primitive is already the raw 6D end-effector pose, together with the gripper state they should represent all possible states of the manipulator. If the primitive cannot be finished within "Hk" steps, one can simply resume the same inputs again. I think more clarifications are needed here.

(2) The parameterization of the primitives. The authors mentioned that the arguments of the primitives can be sampled from a learned distribution. However, the dimension of each primitive's arguments is still different, even if one uses a multi-dimensional Gaussian. More discussions are needed to clarify how arguments are encoded in this scenario.

(3) Choices of primitives. I think a detailed discussion on the importance of each primitive is needed. For example, in order to solve a single task, how many primitives are effectively used? What is the usage percentage/activation time for each of them? Also, a bar plot showing how each primitive is selected over time would be interesting.

(4) On top of (3), one may perform an ablation study on the minimum set of primitives needed to solve a particular task effectively.

(5) Cross robot transfer. These primitives used in appendix seem to be agnostic to robot geometry (unlike joint pose spec). Can we demonstrate how well a policy trained with one robot can be transferred to another robot that is morphologically different?

**Time Spent Reviewing:**

2 hours

---

> ### Author Response · Authors · 2021-08-10
> **Response to Reviewer WUWx**
>
> We thank the reviewer for their valuable comments and suggestions.
> > *(1) If the dummy primitive is already the raw 6D end-effector pose, together with the gripper state they should represent all possible states of the manipulator. If the primitive cannot be finished within "Hk" steps, one can simply resume the same inputs again. I think more clarifications are needed here.*
>
> - Since the primitive is given a fixed goal for H_k steps, it is less expressive than a feedback policy that could provide a changing argument at every low-level step. For example, if the task is to move in a circle, the dummy primitive with a fixed argument could not provide a target state that would directly result in the desired motion without resorting to a significant number of higher level actions, while a feedback policy could iteratively update the target state to produce a smooth motion in a circle. We will clarify this point in the paper.
>
>
>
> > *(2) “More discussions are needed to clarify how arguments are encoded in this scenario.”*
>
> - We provide a discussion on how the arguments are outputted by the higher level policy in paragraph 4 of Section 3.2 in the main text. To provide further intuition, we describe a concrete example here. Let's say if we have 10 primitives with 3 arguments each, the higher level policy network outputs 30 dimensional mean and standard deviation vectors from which we sample a 30 dimensional argument vector. It also outputs a 10 dimensional logit vector from which we sample a 10 dimensional one-hot vector. Therefore in total, our action space would be 40 dimensional. The environment takes in the 40 dimensional vector and selects the appropriate argument (3-dim vector) from the argument vector based on the one-hot vector over primitives and executes the corresponding primitive in the environment. We will add this example to the paper to improve clarity and release the code reproducing experimental results in the paper.
>
>
>
> > *(3) “how many primitives are effectively used? What is the usage percentage/activation time for each of them?”*
>
> - We provide this here: [https://drive.google.com/file/d/1T1YfEWzT7nV5aD4Q2iqiGAgMBigTyiDz/view](https://drive.google.com/file/d/1T1YfEWzT7nV5aD4Q2iqiGAgMBigTyiDz/view)
>
> - In the link above, we include a figure logging the number of times each primitive is called at test time, averaged across all of the kitchen environments. It is clear from the figure that even at convergence, each primitive is called a non-zero amount of times. However, there are two primitives that are favored across all the tasks, move_delta_ee_pose and angled_xy_grasp. This is not surprising as these two primitives are easily applicable to many tasks. We additionally plot the number of unique primitives selected by the test time policy over time (within a single episode) and note that it converges to about 2.69. To ground this number, the path length for these tasks is 5. This means that on most tasks, the higher level policy ends up repeatedly applying certain primitives in order to achieve the task.
>
>
>
> > *(4) “one may perform an ablation study on the minimum set of primitives needed to solve a particular task effectively.”*
>
> - This is an interesting ablation to perform, and we had thought about attempting this experiment in our original submission. However, in general, this set is intractable to estimate. With 10 primitives we would need $\sum_{i=1}^{9} {10 \choose i}$ experiments. More importantly, this is contrary to the paper’s main goal which is to minimize the human effort in the training process by sharing the same superset of primitives across all tasks. Hence, instead of manually picking primitives for each new task, the learning method should learn to automatically discard the non-useful primitives from the superset. Across all the experiments, we found this to be the case.
>
> - Upon suggestion from R3 # mpsj, we also showed that the same set of primitives are able to fully solve **43/50 tasks** in Meta-World suite: *[https://drive.google.com/file/d/1jvKZyUEjhcbVGKoS-UvmNFsSuTBelQEX/view](https://drive.google.com/file/d/1jvKZyUEjhcbVGKoS-UvmNFsSuTBelQEX/view)
>
> > *(5) “Can we demonstrate how well a policy trained with one robot can be transferred to another robot that is morphologically different?”*
>
> - We trained an RL agent using our method to solve the Robosuite door opening task using the xArm 7. This agent is able to achieve a 100% success rate after about 10 hours of training. We then evaluate this agent, zero-shot on the same task but with a different robot, the xArm 6. The agent achieves a **100% success rate after transfer** to the xArm 6. While xArm 7 is a 7 DOF manipulator, the xArm 6 has only 6 degrees of freedom, so the two robots have a different morphology. This demonstrates the capability of our proposed method to automatically generalize between robots with different morphologies, a property that the raw action space and other skill learning methods do not have. We thank the reviewer for suggesting this experiment and definitely plan to include this new result in the paper.

---

### Author Response · Authors · 2021-08-10
**Short summary of rebuttal to all**

We would like to thank reviewers for their valuable feedback. To summarize, we design a simple yet effective hierarchical action space for robotic manipulation and execute a careful and thorough empirical evaluation demonstrating its effectiveness on a wide array of tasks.



We are pleased to report that we have finished all the additional baselines or experiments suggested by the reviewers. Here is the document listing all of them (anonymized): [https://drive.google.com/file/d/1IjwJ99iNthLTmbVbChq0xQHkFtDR2Ibz/view](https://drive.google.com/file/d/1IjwJ99iNthLTmbVbChq0xQHkFtDR2Ibz/view)

We provide full detail in direct replies to each review. Here, we recap the main points from reviewers and summarize the experimental findings:

-   **Cross robot transfer experiment [R1 # WUWx]:** R1 suggested transferring policies between robots with different morphologies. We confirmed that this is indeed possible: for the door opening task in Robosuite, we trained a policy using the 7 DOF xArm 7 robot and transferred it successfully to the 6 DOF xARM 6 robot. We plan to include this result in the final version of the paper.
-   **Dynamic movement primitives (DMP) [R2 # cseK, R3 # mpsj]:** R2 and R3 note the connections between our method and movement primitives, a connection which we also had recognized in our related work. Based on the reviewers’ feedback, we include a comparison to a DMP-based RL method. We found that RAPS (our method) significantly outperforms NDP across nearly every task in the Kitchen suite. Please see the reply to R2 for more details.
-   **Potentially only applicable to a limited set of tasks? [R3 # mpsj]:** We demonstrate that RAPS scales to a significantly larger array of tasks than originally evaluated in the submission. Specifically, using RAPS, we are able to **fully solve 43/50 tasks in the Meta-world suite** from image input with sparse rewards. We provide these results in the pdf link above and will include them in the paper.
- **Low-level actions are the bottleneck in the real world. [R3 #mpsj]:** We ran an experiment in the real world to test this hypothesis. We found that, per episode, on a real xArm 6 robot, RAPS is **32x** faster than low-level control, when used inside an RL loop. We discuss this result in more detail in the reply to R3 and will include this in an updated version of the paper.

---

### Decision · Program_Chairs · 2021-09-27

**Decision:**

Accept (Poster)

**Comment:**

The reviewers had some concerns that were all addressed by the authors. Especially, the authors did new experiments and provided a lot of new results. All these new results are extremely useful to support the claims of the authors and *should be included* in the final version of the paper. For this reason, the paper will have to be significantly revised but, on the basis of the discussion, the reviewers are confident that this final version will describe a strong enough contribution.